# System-1.5 Reasoning: Traversal in Language and Latent Spaces with Dynamic Shortcuts

**Xiaoqiang Wang**[1,2]    **Suyuchen Wang**[1,2]    **Yun Zhu**   **Bang Liu**[1,2,3†]

[1]DIRO & Institut Courtois, Université de Montréal
[2]Mila - Quebec AI Institute; [3]Canada CIFAR AI Chair
{xiaoqiang.wang, suyuchen.wang, bang.liu}@umontreal.ca
gabrielzhuyun@gmail.com

## Abstract

Chain-of-thought (CoT) reasoning enables large language models (LLMs) to move beyond fast System-1 responses and engage in deliberative System-2 reasoning. However, this comes at the cost of significant inefficiency due to verbose intermediate output. Recent latent-space reasoning methods improve efficiency by operating on hidden states without decoding into language, yet they treat all steps uniformly, failing to distinguish critical deductions from auxiliary steps and resulting in suboptimal use of computational resources. In this paper, we propose `System-1.5 Reasoning`, an adaptive reasoning framework that dynamically allocates computation across reasoning steps through shortcut paths in latent space. Specifically, `System-1.5 Reasoning` introduces two types of dynamic shortcuts. The model depth shortcut (DS) adaptively reasons along the vertical depth by early exiting non-critical tokens through lightweight adapter branches, while allowing critical tokens to continue through deeper Transformer layers. The step shortcut (SS) reuses hidden states across the decoding steps to skip trivial steps and reason horizontally in latent space. Training `System-1.5 Reasoning` involves a two-stage self-distillation process: first distilling natural language CoT into latent-space continuous thought, and then distilling full-path System-2 latent reasoning into adaptive shortcut paths (`System-1.5 Reasoning`). Experiments on reasoning tasks demonstrate the superior performance of our method. For example, on GSM8K, `System-1.5 Reasoning` achieves reasoning performance comparable to traditional CoT fine-tuning methods while accelerating inference by over 20× and reducing token generation by 91.0% on average.

## 1 Introduction

Foundational large language models (LLMs) (Ouyang et al., 2022; Team et al., 2023; Dubey et al., 2024; Hurst et al., 2024; Meta, 2025) has revolutionized natural language processing (Liang et al., 2022; Srivastava et al., 2023; Wang et al., 2024) and demonstrated strong potential in diverse agent applications (Liu et al., 2025), automating real-world tasks across both digital (OpenAI, 2022; Wang et al., 2023; Hong et al., 2023; Wang & Liu, 2024; Qin et al., 2025; Wang et al., 2025) and physical environments (Brohan et al., 2023; Driess et al., 2023; Zheng et al., 2024; Team et al., 2024). Recent advances in test-time scaling paradigms (Snell et al., 2024; Zhang et al., 2025) and the emergence of reasoning-oriented LLMs, also referred to as Large Reasoning Models (LRMs) (Jaech et al., 2024; Guo et al., 2025; Team, 2025; Abdin et al., 2025), have leveraged Chain-of-Thought (CoT) (Wei et al., 2022) to extend LLM reasoning capabilities (Chen et al., 2025; Wu et al., 2025). This approach facilitates a transition from fast and heuristic-driven System-1 reasoning to slower and deliberate

---

[†]Corresponding author.

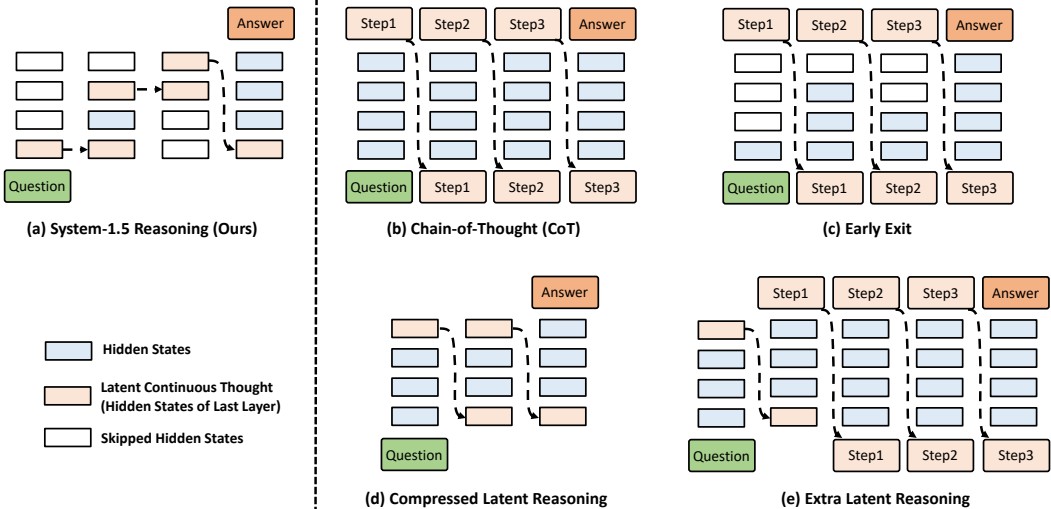

Figure 1: Comparison of (a) our proposed `System-1.5 Reasoning`, (b) chain-of-thought (CoT) reasoning, (c) early-exit (Elbayad et al., 2019; Elhoushi et al., 2024), (d) compressed latent-space reasoning (*e.g.*, Coconut (Hao et al., 2024) and CCoT (Cheng & Van Durme, 2024)), and (e) extra latent reasoning approaches that delay output (*e.g.*, *pause* token (Goyal et al., 2023) and *filler* token (Pfau et al., 2024)). `System-1.5 Reasoning` enables flexible latent reasoning along both the horizontal and vertical dimensions through dynamic shortcuts.

System-2 reasoning (Li et al., 2025b), and has significantly improved performance in competitive mathematics (Hendrycks et al., 2021; Li et al., 2024) and code generation (Jimenez et al., 2023; Jain et al., 2024).

However, CoT reasoning incurs substantial computational costs during inference due to the need to generate long reasoning chains before producing the final answer. It also suffers from overthinking phenomena (Chen et al., 2024; Team et al., 2025), resulting in redundant and verbose outputs even for simple problems, where excessive reasoning leads to unnecessary token generation and escalates inefficiency.

To address it, one line of work focuses on language-space efficiency, such as imposing token budgets during prompt input (Xu et al., 2025a) or decoding interventions (Muennighoff et al., 2025) to constrain the length or complexity (Lee et al., 2025) of generated explanations. Another direction emphasizes efficient decoding, such as speculative decoding (Leviathan et al., 2023) and optimized sampling (Sun et al., 2024b; Fu et al., 2024; Li et al., 2025a).

However, many tokens generated during chain-of-thought (CoT) reasoning primarily serve textual fluency and coherence rather than contributing meaningfully to actual reasoning advancement (Song et al., 2025; Luo et al., 2025a). Consequently, recent approaches have explored operating reasoning in latent space. Methods such as the *pause* token (Goyal et al., 2023) and *filler* token (Pfau et al., 2024) insert special tokens and ignore their language-space outputs to allow for additional latent computation, while Coconut (Hao et al., 2024) and CCoT (Cheng & Van Durme, 2024) compress explicit CoT into hidden states and feed these hidden states, rather than generated tokens, back into the model as subsequent embeddings. By bypassing language constraints, these internal computations enable more compact and flexible reasoning within the latent space.

While these methods improve efficiency, as illustrated in Figure 1, they introduce a unidirectional optimization bias, promoting either universally fast reasoning through compressed latent-space reasoning (*e.g.*, Coconut and CCoT) for all reasoning steps or universally slow reasoning that delays outputs at every step (*e.g.*, *pause* token and *filler* token). This uniform treatment fails to differentiate between critical deductions and auxiliary steps, leading to inefficient allocation where trivial and complex steps receive nearly equal computational budgets. For example, a CoT sequence often includes distinct phases such as problem restatement, exploration, and result verification, each requiring different levels of reasoning effort (Luo et al., 2025b).

This raises a natural question: *Can we dynamically tailor computation across different reasoning steps to maximize efficiency while preserving performance?* Ideally, models should reason quickly (System-1) on non-critical steps and more carefully (System-2) on critical ones.

Motivated by this, we propose `System-1.5 Reasoning`, which adaptively combines fast and slow reasoning paths in latent space to handle varying step complexities in the language space. `System-1.5 Reasoning` introduces model depth shortcut (DS) and decoding step shortcut (SS) to adaptively allocate computation along the vertical and horizontal paths in latent space.

Specifically, the depth shortcut is implemented by augmenting each Transformer layer with a router-adapter module, which dynamically determines whether a token continues through deeper standard layers or exits early via a lightweight adapter branch. This depth shortcut mechanism enables flexible vertical computation along the model depth, allowing non-critical reasoning steps to be processed with fewer layers while critical steps continue deeper into the model. In addition, the step shortcut enables latent-space skipping across the horizontal dimension of decoding length, where early-exited hidden states at a given layer are directly copied to the next decoding step, instead of restarting latent computation from the first layer as in standard Transformers.

By supporting adaptive reasoning along both vertical and horizontal paths, `System-1.5 Reasoning` maximizes flexibility in latent-space reasoning and better mirrors human thinking, where difficult reasoning steps are handled with deliberate, reflective System-2 thinking, simple steps are processed quickly through heuristic System-1 thinking, and trivial steps are naturally skipped without thinking.

To enable latent-space shortcut reasoning, we train `System-1.5 Reasoning` via a two-stage distillation process. The first stage, language-to-latent distillation, aligns the latent-space behavior of a student model with the language-space CoT reasoning of a teacher model. This process leverages full teacher-forcing and allows efficient parallel training, avoiding the step-wise scheduling complexities of curriculum learning approaches such as iCoT (Deng et al., 2024) and Coconut (Hao et al., 2024).

The second stage, System-2 to System-1.5 distillation, further compresses full-depth reasoning trajectories into shortcut execution path by leveraging reasoning criticality estimation in the language space. Specifically, we freeze the original transformer parameters and tune only the router-adapter module using an early-exit loss that encourages non-critical tokens to exit at earlier layers and critical tokens to proceed to deeper layers, while maintaining consistency between the hidden states of the full path and those of the shortcut-exited path.

We validate our approach on challenging reasoning datasets such as GSM8K (Cobbe et al., 2021). Experiments demonstrate that `System-1.5 Reasoning` achieves reasoning performance comparable to traditional CoT fine-tuning methods while accelerating inference by over $20\times$ and reducing token generation by 91.0% on average. These results highlight the promise of `System-1.5 Reasoning` in improving the efficiency and scalability of the LLM reasoning.

## 2  System-1.5 Reasoning

We frame `System-1.5 Reasoning` as adaptive and dynamic latent-space reasoning guided by criticality analysis of language-space reasoning. As shown in Figure 2, the core idea is to dynamically allocate computation in latent space through two types of **dynamic shortcuts**, model depth shortcut (DS) and decoding step shortcut (SS), to handle varying step complexities in the language space. Specifically, by inserting a standard router-adapter module into each vanilla Transformer layer, the dynamic depth shortcut enables simple steps to be processed through shallow layers, while complex steps are routed through deeper layers for more extensive computation. In parallel, the dynamic step shortcut allows trivial steps to be skipped by copying hidden states at early exit points and directly reusing them as the hidden states for the next decoding step at the same layer.

To train `System-1.5 Reasoning` effectively in latent space, we employ a **two-stage distillation** process. First, we perform language-to-latent distillation by fine-tuning vanilla Transformer layers, aligning the student model's hidden states, which reason in latent space with those of the CoT-trained teacher model. Then, with the vanilla Transformer parameters frozen, we perform System-2 to System-1.5 distillation by training the router-adapter modules. Specifically, we leverage atomic thought decomposition (Teng et al., 2025) to estimate reasoning criticality step by step. We decompose the CoT into a directed acyclic graph (DAG) of self-contained subquestions. In this graph, independent nodes are labeled as non-critical, while derived nodes that require logical integration are labeled as

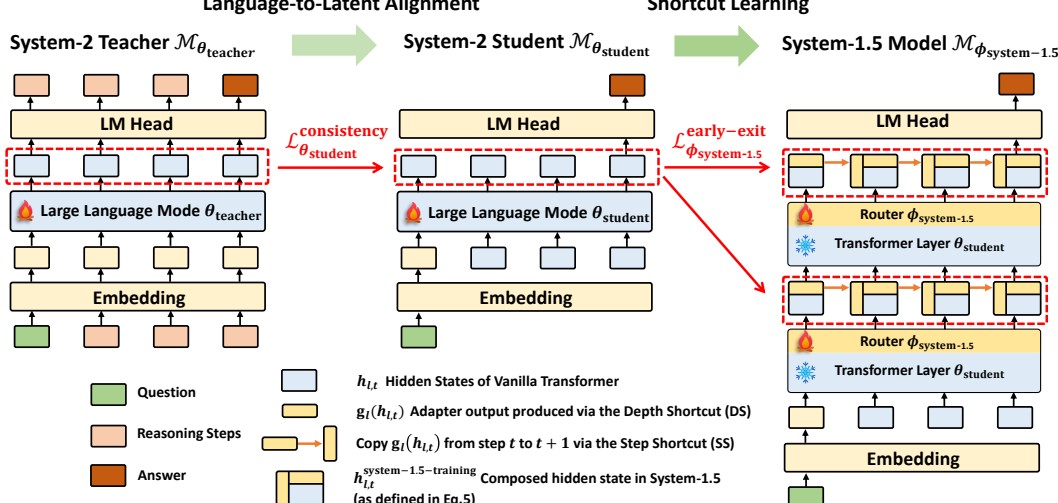

Figure 2: `System-1.5 Reasoning` is trained through a two-stage distillation process: (1) Language-to-latent alignment, where the model learns to reason in latent space by minimizing a consistency loss between a language-space reasoning model (System-2 teacher) and a latent-space reasoning model (System-2 student); and (2) Shortcut learning, where dynamic shortcuts are trained by applying an early-exit loss that encourages non-critical steps to exit earlier and critical steps to proceed deeper. The model is simplified to a 2-layer Transformer for illustration purposes.

critical. Based on this, we apply an early-exit loss that encourages tokens of non-critical steps to exit earlier and tokens of critical steps to proceed to deeper layers.

## 2.1 Dynamic Shortcut Architecture

Formally, given an input text sequence $X = \langle x_1, \cdots, x_t, \cdots, x_T \rangle$, we denote the vanilla hidden states at the $l$-th layer as $H_l = \langle h_{l,1}, \cdots, h_{l,t}, \cdots, h_{l,T} \rangle$, where $T$ is the sequence length and $l \in \{0, 1, \cdots, L\}$ with $L$ being the total number of Transformer layers. The initial hidden states are given by $H_0 = \text{Embed}(X)$, corresponding to the embedding layer. For $l \geq 1$, denoting $f_l(\cdot)$ as the operation of a single Transformer layer, the hidden state at layer $l$ is updated as $H_l = f_l(H_{l-1})$.

**Depth shortcut.** To adaptively determine how many of the initial Transformer layers are needed for each token, *i.e.*, when to trigger early exit, we employ a lightweight router-adapter module, as shown in Figure 2. At each Transformer layer $l$, the router module $\mathcal{R}_l$ dynamically decides whether a token should continue through the standard Transformer layer $f_l$ for deeper processing, or exit early through an adapter branch $g_l$. Formally, during training, the output at layer $l$ for step $t$ is expressed as a weighted combination of the adapter and Transformer outputs, where the weight is determined by the binary router $\mathcal{R}_l$, implemented as a feed-forward network (FFN) layer followed by a sigmoid activation:

$$h_{l,t}^{\text{system-1.5-ds-training}} = g_{l-1}(h_{l-1,t}) * w + f_l(h_{l-1,t}) * (1 - w) \tag{1}$$

$$w = \mathcal{R}_l(h_{l-1,t}) \tag{2}$$

During inference, the output of the router $\mathcal{R}l$ serves as a confidence score for early exit, which is compared against a predefined depth exit threshold $\lambda_{\text{depth}}$ to determine whether to halt computation at the current layer for the given decoding step.

$$h_{l,t}^{\text{system-1.5-ds-inference}} = \begin{cases} g_{l-1}(h_{l-1,t}), & \text{if } \mathcal{R}_l(h_{l-1,t}) > \lambda_{\text{depth}}, \\ f_l(h_{l-1,t}), & \text{otherwise.} \end{cases} \tag{3}$$

**Step shortcut.** The step shortcut is motivated by the observation that, in standard decoding, each step still forces the model to process from the first layer. In other words, while depth shortcuts allow adaptive reasoning along the vertical path in latent space, enabling exits at intermediate layers rather than always reaching the final layer, the model must still sequentially process every decoding step. To address this, we introduce dynamic shortcuts along the decoding steps, allowing the model

to adaptively skip steps and reason horizontally in latent space. By supporting adaptive reasoning along both vertical and horizontal paths, this design maximizes flexibility in latent-space reasoning and enables `System-1.5 Reasoning` to better mirror human thinking, where trivial steps are often skipped (step shortcut), difficult steps are handled with deliberate and reflective reasoning, and simple steps are processed quickly through heuristic reasoning (depth shortcut). More importantly, since `System-1.5 Reasoning` operates entirely in latent space, where new decoding steps rely solely on the hidden states from the previous step without introducing new input language tokens, the practice of step shortcut avoids the problem of partial observation of the input context (*i.e.*, skipping token input at the current step) in token compression methods. This allows `System-1.5 Reasoning` to achieve high computational efficiency while preserving all crucial information and thinking steps.

Formally, similar to the depth shortcut, during training, the output at layer $l$ and step $t$ is computed as a weighted combination of the hidden state from step $t-1$ at the same layer (note that the step shortcut is not applied when $t = 0$) and the hidden state at step $t$ from the vanilla Transformer:

$$h_{l,t}^{\text{system-1.5-ss-training}} = g_l(h_{l,t-1}) * w + f_l(h_{l-1,t}) * (1 - w) \tag{4}$$

where the weight $w$ is determined by the router $\mathcal{R}_l$ as Eq. 2. By merging Eq. 1 and Eq. 4, we obtain:

$$h_{l,t}^{\text{system-1.5-training}} = \Big( \underbrace{g_{l-1}(h_{l-1,t})}_{\text{depth shortcut}} + \underbrace{g_l(h_{l,t-1})}_{\text{step shortcut}} \Big) * w + \underbrace{f_l(h_{l-1,t})}_{\text{vanilla path}} * (1 - w) \tag{5}$$

During inference, as determined by Eq. 3, if a decoding step halts computation at an intermediate layer, the hidden state at that layer is directly passed to the next decoding step within the same layer to continue reasoning. If computation halts at the final layer (*i.e.*, after traversing all the Transformer layers via dynamic shortcuts, termed as a reasoning *cycle*), we either (1) feed the final hidden state back for the next reasoning cycle or (2) generate the final output. Borrowing the latent reasoning setting from (Hao et al., 2024), we apply a fixed number of latent reasoning steps, denoted as the decoding step constant $\lambda_{\text{step}}$. By combining depth exit threshold $\lambda_{\text{depth}}$ in Eq. 3 and decoding step constant $\lambda_{\text{step}}$ to control computation along both the model depth and decoding steps, this approach provides a flexible computational budget for test-time scaling (see Section 3.2).

### 2.2 Two-Stage Distillation: Language-to-Latent Alignment and Shortcut Learning

Training a model to reason in latent space with shortcut paths poses two challenges: the latent property and the adaptive property. (1) Vanilla Transformer layers in pre-trained LLMs are optimized for next-token prediction in the language space and are not inherently capable of reasoning in latent space. How can we align latent-space reasoning with language-space CoT? (2) Adaptive reasoning needs the model to dynamically decide varying reasoning paths in the latent space. How can we potentially leverage the characteristics of language-space reasoning steps (*e.g.* reasoning step criticality) to guide the model toward learning adaptive reasoning via shortcut paths?

As shown in Figure 2, for the latent property challenge, we employ **language-to-latent alignment** through hidden-state distillation. Specifically, we align the reasoning processes between a language-space teacher model and a latent-space student model. The teacher model, denoted as $\mathcal{M}_{\theta_{\text{teacher}}}$, is trained with standard CoT fine-tuning to generate both intermediate steps and final answers in the language space. The student model, denoted as $\mathcal{M}_{\theta_{\text{student}}}$, learns to reason directly in latent space.

For the adaptive property challenge involving dynamic shortcut decisions, we leverage atom-of-thought (Teng et al., 2025) to decompose the original CoT into a directed acyclic graph (DAG) of self-contained subquestions. Independent nodes are identified as non-critical steps (requiring less computation), while derived nodes requiring logical integration are identified as critical steps (requiring deeper reasoning). Based on this, we introduce **shortcut learning**: We first initialize the System-1.5 model, denoted as $\mathcal{M}_{\phi_{\text{system-1.5}}}$, by inheriting the parameters from $\mathcal{M}_{\theta_{\text{student}}}$. We then freeze the Transformer parameters and insert an adapter-router module into each Transformer layer to enable fine-tuning for adaptive reasoning. Lastly, we apply an early-exit loss that encourages non-critical steps to exit at shallower layers while allowing critical steps to proceed deeper into the model.

**Language-to-latent alignment.** During training, we extract the last-layer hidden states from the teacher model, apply stop-gradient, and use them as ground-truth features for the student model. These

features are then fed into the student model together with the question input, enabling teacher-forcing and ensuring training efficiency.

Formally, given an input sequence $X = \langle x_1, \ldots, x_T \rangle$, we further represent it as the concatenation of three segments, the question $Q$, intermediate reasoning steps $R$, and the answer $A$, denoted as $X = \langle Q : R : A \rangle = \langle q_1, \ldots, q_{T_Q}, r_1, \ldots, r_{T_R}, a_1, \ldots, a_{T_A} \rangle$, where $T_Q$, $T_R$, and $T_A$ correspond to the lengths of each segment.

The last-layer hidden states from the System-2 teacher and student models, denoted by $h_{L,t}^{\text{teacher}}$ and $h_{L,t}^{\text{student}}$ respectively, are aligned by minimizing the following mean squared error (MSE) loss:

$$\mathcal{L}_{\theta_{\text{student}}}^{\text{consistency}} = \frac{1}{T_A} \sum_{t=T_Q+T_R+1}^{T_Q+T_R+T_A} \text{MSE}(\text{sg}[h_{L,t}^{\text{teacher}}], h_{L,t}^{\text{student}}) \tag{6}$$

where $\text{sg}[\cdot]$ denotes the stop-gradient operation applied to the teacher model.

In parallel, the System-2 teacher model is supervised by the standard negative log-likelihood (NLL) loss over both intermediate reasoning steps and final answers, while the System-2 student model is supervised by NLL loss over final answer generation only (*i.e.*, performing latent reasoning for intermediate steps):

$$\mathcal{L}_{\theta_{\text{teacher}}}^{LM} = -\sum_{t=T_Q+1}^{T_Q+T_R} \log \mathcal{M}_{\theta_{\text{teacher}}} (r_t \mid r_{<t}, Q) - \sum_{t=T_Q+T_R+1}^{T_Q+T_R+T_A} \log \mathcal{M}_{\theta_{\text{teacher}}} (a_t \mid a_{<t}, R, Q) \tag{7}$$

$$\mathcal{L}_{\theta_{\text{student}}}^{LM} = -\sum_{t=T_Q+T_R+1}^{T_Q+T_R+T_A} \log \mathcal{M}_{\theta_{\text{student}}} (a_t \mid a_{<t}, H_L^{\text{teacher}}, Q) \tag{8}$$

where $H_L^{\text{teacher}} = \{h_{L,T_Q}^{\text{teacher}}, h_{L,T_Q+1}^{\text{teacher}}, \cdots, h_{L,T_Q+T_R-1}^{\text{teacher}}\}$ denotes the extracted last-layer hidden states of the reasoning steps, provided as input to the student model to enable teacher-forcing. The overall loss for language-to-latent alignment is thus:

$$\mathcal{L}_1 = \mathcal{L}_{\theta_{\text{teacher}}}^{LM} + \mathcal{L}_{\theta_{\text{student}}}^{LM} + \alpha \mathcal{L}_{\theta_{\text{student}}}^{\text{consistency}} \tag{9}$$

where $\theta_{\text{teacher}}$ and $\theta_{\text{student}}$ refer to the original Transformer parameters of the System-2 teacher and System-2 student models, respectively.

**Shortcut learning.** Given a dataset with labeled intermediate reasoning steps $R = \langle r_1, r_2, \cdots, r_{T_R} \rangle$ and estimated criticality binary labels for each step $(r_1, c_1), \ldots, (r_{T_R}, c_{T_R})$, we define an early-exit loss (Elbayad et al., 2019; Elhoushi et al., 2024) that enforces consistency between the intermediate hidden states of the System-1.5 model and the final-layer hidden states of the System-2 student model (which has been trained through language-to-latent alignment):

$$\mathcal{L}_{\phi_{\text{system-1.5}}}^{\text{early-exit}} = \sum_{l=1}^{L} \sum_{t=T_Q+1}^{T_Q+T_R} e_{l,t} \text{MSE}(\text{sg}[h_{l,t}^{\text{student}}], h_{l,t}^{\text{system-1.5-training}}) \tag{10}$$

where $e_{l,t}$ is the early-exit weight for the $l$-th layer at step $t$, designed to distinguish critical and non-critical steps. Specifically, we define $e_{l,t}$ to encourage non-critical steps ($c_t = 0$) to exit earlier, and critical steps ($c_t = 1$) to exit later, according to:

$$e_{l,t} = (1 - c_t) \frac{\sum_{i=1}^{l} i}{\sum_{j=1}^{L} j} + c_t \left(1 - \frac{\sum_{i=1}^{l} i}{\sum_{j=1}^{L} j}\right). \tag{11}$$

In parallel, the System-1.5 model is also supervised by a NLL loss over the final answer generation:

$$\mathcal{L}_{\phi_{\text{system-1.5}}}^{LM} = -\sum_{t=T_Q+T_R+1}^{T_Q+T_R+T_A} \log \mathcal{M}_{\phi_{\text{system-1.5}}} (a_t \mid a_{<t}, H_L^{\text{student}}, Q) \tag{12}$$

Table 1: Quantitative results on reasoning tasks, reported in terms of accuracy (**Acc.**), number of decoding steps before generating the final answer (**# Steps**), average FLOPs reduction rate per step relative to CoT (**FLOPs r.**), and overall inference speedup relative to CoT measured by wall-clock time. Best and second-best results are highlighted with **bold** and underline, respectively.

| Method | GSM8K | | | | GSM-HARD | | | | StrategyQA | | | |
|---|---|---|---|---|---|---|---|---|---|---|---|---|
| | Acc. (%) | # Steps | FLOPs r. | Speedup | Acc. (%) | # Steps | FLOPs r. | Speedup | Acc. (%) | # Steps | FLOPs r. | Speedup |
| CoT | **46.94** | 26 | - | - | **38.32** | 26 | - | - | 47.62 | 52 | - | - |
| LITE | 44.51 | 28 | 1.84× | 1.61× | 37.01 | 28 | 1.56× | 1.44× | 46.15 | 42 | 1.96× | 2.36× |
| LayerSkip | 43.20 | 32 | 1.77× | 1.45× | 36.55 | 33 | 1.32× | 1.03× | 42.54 | 49 | 1.8× | 1.86× |
| iCoT | 32.14 | 2 | 1.02× | 13.45× | 23.17 | 4 | 1.02× | 6.17× | 34.42 | 2 | 1.02× | 26.47× |
| Coconut | 36.75 | 2 | 1.02× | 11.98× | 28.25 | 4 | 1.02× | 7.07× | 38.67 | 2 | 1.02× | 27.25× |
| CODI | 43.78 | 2 | 1.02× | 13.37× | 35.91 | 4 | 1.02× | 6.93× | 45.12 | 2 | 1.02× | 25.96× |
| *pause* token | 46.32 | 38 | 1.00× | 0.8× | 38.17 | 42 | 1.00× | 0.89× | 48.14 | 61 | 1.00× | 0.82× |
| System-1.5 | 46.66 | 2 | 1.95× | **20.27×** | 38.28 | 4 | 1.76× | **12.45×** | **48.61** | 2 | 2.12× | **55.65×** |

where $H_L^{\text{student}}$ extracts the System-2 student model to enable teacher-forcing. The overall loss for shortcut learning is then given by:

$$\mathcal{L}_2 = \mathcal{L}_{\phi_{\text{system-1.5}}}^{LM} + \beta \mathcal{L}_{\phi_{\text{system-1.5}}}^{\text{early-exit}} \tag{13}$$

where $\phi_{\text{system-1.5}}$ refers to the parameters of the router-adapter modules. The underlying Transformer parameters are initialized from $\theta_{\text{student}}$ and are kept frozen during shortcut learning.

# 3 Experiments

**Datasets.** We evaluate the effectiveness and efficiency of `System-1.5 Reasoning` on two reasoning-intensive tasks: mathematical reasoning and common sense reasoning. For mathematical reasoning, we train on the augmented GSM8K dataset (Deng et al., 2023), which extends the original GSM8K (Cobbe et al., 2021) with a larger set of grade school-level math problems. For common-sense reasoning, we use StrategyQA (Geva et al., 2021), which contains multihop questions annotated with supporting facts. Each sample includes a question, decomposed subquestions, and a yes/no answer. We merge the annotated facts and subquestions as a coherent CoT sequence for training. For both datasets, we label each reasoning step with criticality annotations using atomic thought decomposition, as described in Section 2.2, to distinguish between critical and non-critical steps. We train models on the official training splits and evaluate performance on the respective test sets for in-domain evaluation. Additionally, for mathematical reasoning, we conduct out-of-domain evaluation on GSM-HARD (Gao et al., 2023), a dataset with increased reasoning difficulty designed to test generalization beyond the original GSM8K.

**Baselines.** In addition to CoT fine-tuning, we compare `System-1.5 Reasoning` against six efficient reasoning methods based on supervised fine-tuning, consisting of: (1) language-space conditional computation methods: LITE (Varshney et al., 2023) and LayerSkip (Elhoushi et al., 2024), which improve efficiency by applying early exit mechanisms; (2) Latent-space compressed reasoning: iCoT (Deng et al., 2024), Coconut (Hao et al., 2024), and CODI (Shen et al., 2025), which compress natural language CoT into compact continuous latent representations; (3) Latent-space extended reasoning methods: *pause* token (Goyal et al., 2023), which insert extra latent states to delay output and allow for extended internal reasoning. Since the official implementations of iCoT and Coconut are based on GPT-2 124M (Radford et al., 2019), while LayerSkip builds on the LLaMA 2 (Touvron et al., 2023) and LLaMA 3 (Grattafiori et al., 2024) series model, we implement two backbone versions of `System-1.5 Reasoning`, one using GPT-2 124M and another using LLaMA 3.2 1B, to ensure fair comparisons between different baselines.

## 3.1 Main Results

`System-1.5 Reasoning` **outperforms previous state-of-the-art methods in latent-space reasoning in both accuracy and efficiency.** As shown in Table 1, compared to iCoT, Coconut, CODI, and *pause* token, `System-1.5 Reasoning` achieves higher accuracy and greater overall speedup. In GSM8K, compared to original CoT reasoning, latent-space reasoning reduces intermediate token generation by 92.31% during inference. This reduction is even more pronounced on StrategyQA, reaching 96.15%. Additionally, compared to early-exit methods, `System-1.5 Reasoning` achieves a 1.95× reduction in average FLOPs per decoding step, due to its dynamic step shortcut mechanism,

which allows hidden states from early-exit layers to be directly copied and reused in the next decoding step. In contrast, early-exit methods still reprocess these states starting from the first layer.

By combining early exits along the model depth and shortcut copying across decoding steps, `System-1.5 Reasoning` achieves an overall inference speedup of 20.27× on GSM8K, further improving upon the approximately 10× speedups achieved by previous latent reasoning methods. On StrategyQA, it further accelerates inference by over 55×, significantly enhancing reasoning efficiency.

`System-1.5 Reasoning` **matches CoT fine-tuning on challenging mathematical reasoning tasks and outperforms it on text-rich commonsense reasoning.** On GSM8K and GSM-HARD, `System-1.5 Reasoning` achieves 46.94% and 38.32% accuracy respectively, closely matching CoT fine-tuning results of 46.67% and 38.28%. On StrategyQA, a task requiring reasoning over multiple pieces of textual evidence, `System-1.5 Reasoning` achieves 48.61% accuracy, outperforming CoT's 47.36%. Similar improvements are observed for latent-space reasoning methods such as *pause* token, highlighting the promising potential of `System-1.5 Reasoning` in reducing dependency on explicit textual tokens and enhancing the model's inherent reasoning capabilities.

## 3.2 In-depth Analysis

**Direct distillation from language to latent is more effective for shortcut learning in** `System-1.5 Reasoning`**.** We modify the language-to-latent alignment into a curriculum-based learning process, following the approach used in Coconut. This curriculum consists of multiple stages where natural language thoughts are progressively replaced with latent-space thoughts to train the latent reasoning model (analogous to using a fine-tuned Coconut model as the System-2 student). We then apply the same shortcut learning procedure to train `System-1.5 Reasoning`. We follow Coconut's original training schedule, using six epochs in the initial stage and three in each subsequent stage. For comparison, we also evaluate CODI, a distillation-based latent reasoning model that performs competitively in Table 1, as the System-2 student for shortcut learning.

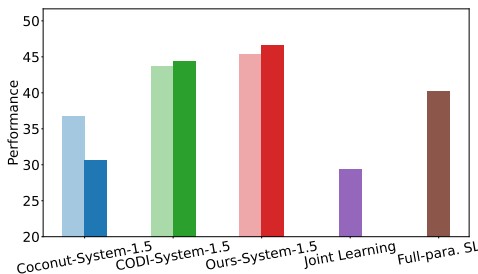

Figure 3: Ablation results on optimizing System-1.5 (shown in solid color) using different System-2 students (shown in light color), namely Coconut-System-1.5 and CODI-System-1.5, as well as alternative learning strategies: joint learning of language-to-latent alignment and shortcut learning, and full-parameter shortcut learning (SL).

Figure 3 shows the results using different System-2 students to train System-1.5 model. We observe a significant accuracy drop in the final `System-1.5 Reasoning` performance when using Coconut as the System-2 student. In addition to Coconut's suboptimal performance observed in Table 1, another possible reason for the gap is that its hard curriculum schedule between language and latent-space reasoning limits the flexibility and completeness of latent-space modeling. In contrast, `System-1.5 Reasoning` leverages both "horizontal" and "vertical" shortcut across decoding steps and model depth, requiring a more flexible latent reasoning structure that hard curriculum distillation cannot effectively support.

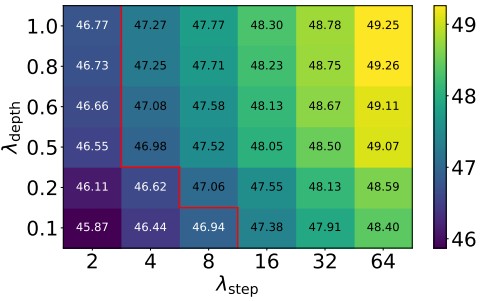

Figure 4: Controllable test-time scaling by tuning the depth exit threshold $\lambda_{\text{depth}}$ and decoding step constant $\lambda_{\text{step}}$ in `System-1.5 Reasoning`. The red grid-aligned Pareto boundary highlighting the frontier of configurations surpassing the CoT baseline (46.94% accuracy in Table 1).

**Joint learning and full-parameter shortcut learning degrade the performance of** `System-1.5 Reasoning`**.** We further investigate the training strategy of `System-1.5 Reasoning` by exploring two variants: (1) Joint learning, where language-to-latent alignment and shortcut learning are conducted simultaneously, *i.e.*, distilling the hidden states of natural language thought directly into the shortcut-exited hidden states; and (2) Full-

parameter shortcut learning, where instead of freezing the original Transformer parameters, all model parameters are updated during shortcut learning.

As shown in Figure 3, both joint learning and full-parameter shortcut learning degrade the final performance of `System-1.5 Reasoning`, with joint learning exhibiting a more pronounced drop. We attribute this to optimization conflicts between the two groups of parameters: Transformer parameters responsible for latent reasoning and router parameters responsible for shortcut routing. These conflicts likely hinder the model's ability to simultaneously preserve flexible latent reasoning paths and optimize dynamic shortcut decisions.

`System-1.5 Reasoning` **enables flexible budget-controllable test-time scaling.** Unlike language-space reasoning, where scaling test-time computation often requires exploring complex structural dependencies (*e.g.*, tree-like exploration) or enforcing solution consistency across multiple trajectories (Zhang et al., 2025), `System-1.5 Reasoning` allows for fine-grained control over computational budgets through simple threshold tuning during inference. Specifically, `System-1.5 Reasoning` inherently introduce two control parameters: the depth exit threshold $\lambda_{\text{depth}}$, which adjusts the adaptive computation depth for each decoding step (vertically across model layers, where $\lambda_{\text{depth}} = 1$ forces the computation to proceed to the final layer as in a standard Transformer), and the decoding step constant $\lambda_{\text{step}}$, which determines when to halt intermediate latent thought generation and output a final answer (horizontally across decoding steps).

Figure 4 shows the performance under different configurations of $\lambda_{\text{depth}}$ and $\lambda_{\text{step}}$. We observe that performance is approximately equally sensitive to adjustments along both dimensions, further validating the motivation for adaptive reasoning across both the model depth and decoding steps. Moreover, depth scaling saturates more quickly, and deeper reasoning demands significantly higher training computation, suggesting a synergistic relationship between train-time scaling and flexible test-time scaling in `System-1.5 Reasoning`.

# 4 Related Works

**Efficient reasoning models.** Efficient reasoning models aim to mitigate the inefficiencies caused by verbose outputs and the overthinking phenomenon (Chen et al., 2024; Team et al., 2025). One line of work focuses on improving language-space efficiency, using length budgeting through prompting (Lee et al., 2025) and fine-tuning (Liu et al., 2024; Yu et al., 2024; Luo et al., 2025a). For example, Chain-of-Draft (Xu et al., 2025a) applies token budget-aware prompting to guide the model toward more concise reasoning, while S1 (Muennighoff et al., 2025) introduces a budget-forcing strategy to terminate the thinking process early. Another line of work focuses on compressing language-space reasoning into latent space, exemplified by iCoT, Coconut (Hao et al., 2024), and CCoT (Cheng & Van Durme, 2024), which train models to reason through compact internal representations. *Pause* token (Goyal et al., 2023) and *filler* token (Pfau et al., 2024) further extend latent-space reasoning by inserting special tokens that delay output and allow for additional latent computation. More recently, Token Assorted (Su et al., 2025) mixes latent and language thoughts by abstracting early steps into discrete latent codes, SoftCoT (Xu et al., 2025b) mitigates catastrophic forgetting in latent reasoning via prompt tuning, and CODI (Shen et al., 2025) applies hidden-state distillation to enhance latent reasoning performance. `System-1.5 Reasoning` builds upon latent-space reasoning but further optimizes efficiency by adaptively allocating computation to handle reasoning steps with varying complexity.

**Conditional computation.** Conditional computation techniques aim to selectively apply heavy computation only to the important parts of an input, thereby reducing unnecessary resource usage. One line of work focuses on system or model switching (Qu et al., 2025), where inputs are routed between different reasoning systems—for example, a fast System-1 and a slower, more deliberate System-2. System-1.x (Saha et al., 2024) combines linear reasoning chains and search trees to enable fast and accurate planning in maze navigation tasks, while FaST (Sun et al., 2024a) uses switch adapters to dynamically toggle between System-1 and System-2 modes based on task complexity factors such as visual uncertainty or occlusion. Another line of work explores sparse activation via mixture-of-experts models (Jiang et al., 2024; Raposo et al., 2024), where only a subset of the model is activated during inference. Beyond expert routing, dynamic depth models adaptively apply only a portion of the Transformer layers. Early exit (EE) and skip-layer techniques are common approaches: DeeBERT (Xin et al., 2020) and CascadeBERT (Li et al., 2021) apply EE in encoder-

only architectures, while more recent decoder-only LLMs like LITE (Varshney et al., 2023) and SkipDecode (Del Corro et al., 2023) employ confidence-based exits to speed up inference. Other approaches, such as CoLT5 (Ainslie et al., 2023) and LayerSkip (Elhoushi et al., 2024), perform binary routing to skip redundant attention layers or entire blocks. `System-1.5 Reasoning` draws inspiration from this line of work and extends it to the latent reasoning space. It not only adapts computation along the model depth via early exits but also introduces dynamic shortcuts across decoding steps, enabling reasoning to proceed efficiently both vertically and horizontally.

## 5    Conclusion

We introduce `System-1.5 Reasoning`, a novel reasoning framework that improves inference efficiency in large language models by enabling dynamic shortcut reasoning within latent space. `System-1.5 Reasoning` adaptively allocates computation based two forms of latent-space shortcuts: depth shortcuts, which modulate layer-wise computation, and step shortcuts, which streamline decoding steps. To support this adaptive reasoning, `System-1.5 Reasoning` is trained via a two-stage distillation process: first aligning language-space and latent-space reasoning, and then learning shortcut execution paths guided by stepwise reasoning criticality. This training paradigm ensures that the model captures both the expressiveness of System-2 reasoning and the efficiency of System-1-style shortcuts. Experimental results on challenging reasoning benchmarks demonstrate that `System-1.5 Reasoning` preserves the accuracy of traditional chain-of-thought methods while accelerating inference by more than $20\times$. These results highlight the potential of `System-1.5 Reasoning` as an effective and scalable latent-space reasoning framework for future LLM deployment.

## Acknowledgements

This work is supported by the Canada CIFAR AI Chair Program and the Canada NSERC Discovery Grant (RGPIN-2021-03115).

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

# A Limitations and Societal Impact

**Limitations.** While `System-1.5 Reasoning` shows promising improvements in efficiency and performance, it comes with two main limitations. First, reasoning in latent space, despite being compact and efficient, lacks the interpretability of language-space CoT methods. Without explicit intermediate reasoning steps, it becomes difficult to understand, analyze, or verify the model's internal logic. Second, our evaluation is currently limited to medium-scale benchmarks and model sizes. Future work is needed to validate `System-1.5 Reasoning` across larger-scale settings and a broader range of tasks.

**Potential Societal Impact.** On the positive side, the efficiency of `System-1.5 Reasoning` could reduce the computational cost of deploying reasoning-capable LLMs, facilitating their broader adoption in real-world applications. This could contribute to more accessible and sustainable AI technologies, potentially democratizing advanced reasoning capabilities for a wider range of users and institutions. On the negative side, the lack of interpretability inherent to latent reasoning poses risks in high-stakes or safety-critical settings. Without transparent intermediate steps, it is harder to detect flawed logic or harmful behavior, which may require stronger safeguards than those typically used for explicit language-based reasoning systems.

# B Experiment Setup

**Datasets.** We evaluate the effectiveness and efficiency of `System-1.5 Reasoning` in two reasoning-intensive tasks: mathematical reasoning and common sense reasoning.

For mathematical reasoning, we utilize the augmented GSM8K dataset (Deng et al., 2023), which extends the original GSM8K dataset (Cobbe et al., 2021) to approximately 400,000 grade school-level math problems. Each problem comprises a question, a chain-of-thought (CoT) that details the intermediate reasoning steps, and a final numerical answer. To assess generalization to more challenging problems, we also perform out-of-domain evaluation using GSM-HARD (Gao et al., 2023), a dataset derived from GSM8K by replacing numerical values with larger, less common numbers, thus increasing the difficulty of the problem.

For commonsense reasoning, we employ the StrategyQA dataset , which consists of 2,780 examples requiring implicit multihop reasoning. Each example includes a concise yes/no question, a decomposition into intermediate reasoning steps, and supporting evidence paragraphs. We construct CoT sequences by merging the annotated decompositions and corresponding evidence, enabling the model to learn structured reasoning paths.

To facilitate adaptive reasoning, we annotate each CoT step in both datasets with criticality labels, identifying steps as either critical or non-critical, using atomic thought decomposition, as described in Section 2.2. For both tasks, we train `System-1.5 Reasoning` on the respective training splits and evaluate performance on the corresponding test sets to assess in-domain effectiveness. The inclusion of GSM-HARD allows us to further evaluate the model's ability to generalize to more complex mathematical reasoning scenarios.

**Baselines.** Beyond standard CoT fine-tuning, we compare `System-1.5 Reasoning` against six efficient supervised fine-tuning-based reasoning methods, categorized into three classes: (1) Language-space conditional computation methods: LITE (Varshney et al., 2023) and LayerSkip (Elhoushi et al., 2024), which improve efficiency by applying early exit mechanisms on decoding natural language thought sequences; (2) Latent-space compressed reasoning methods: iCoT (Deng et al., 2024), Coconut (Hao et al., 2024), and CODI (Shen et al., 2025), which compress natural language CoT into compact continuous latent representations; (3) Latent-space extended reasoning methods: *pause* token (Goyal et al., 2023), which insert extra latent states to delay output and allow for extended internal reasoning.

Since the official implementations of iCoT and Coconut are based on GPT-2 124M, while LayerSkip builds on the LLaMA 2 and LLaMA 3 series, we implement two backbone versions of `System-1.5 Reasoning`, one using GPT-2 124M and another using LLaMA 3.2 1B, to ensure fair comparisons across different baselines.

**Evaluation metrics.** Accuracy is evaluated by exact match between the generated final answer and the ground truth: specifically, the numerical value in the GSM8K dataset, and the yes/no label in

StrategyQA. In addition to accuracy, we assess efficiency across three dimensions: decoding steps, computation cost, and overall inference time.

Decoding steps refer to the number of tokens generated before producing the final answer. For latent reasoning models that do not generate explicit tokens, we instead count the number of latent reasoning steps applied. We unify the terminology across all methods by reporting the average number of decoding steps required on the test set.

The computation cost considers both the number of decoding steps and the computational depth traversed within the Transformer layers. That is, although different methods may require a similar number of decoding steps or generated tokens, methods like `System-1.5 Reasoning` and conditional computation models with early exits involve significantly fewer layer computations. Following the setting in previous work (Elbayad et al., 2019; Schuster et al., 2022), we estimate the computational cost using FLOPs (floating point operations), approximating the dot-product and matrix-vector operations within Transformer layers, while omitting nonlinearities, normalization, and residual connections. We compute the average FLOPs per decoding step and report the average reduction in FLOPs relative to CoT fine-tuning.

The overall inference time is measured by recording the actual wall-clock inference time on the test set. We report the average inference speedup relative to CoT fine-tuning.

**Implementation Details.** We developed our method using PyTorch (Paszke et al., 2019). The base models—GPT-2 124M (Radford et al., 2019) and LLaMA 3.1-1B (Grattafiori et al., 2024)—are initialized from pretrained checkpoints provided by the Hugging Face Transformers library (Wolf et al., 2020). During shortcut learning, we insert a router-adapter module into each Transformer layer. The router module is implemented as a feed-forward network (FFN) followed by a sigmoid activation. The adapter module is implemented using LoRA (Hu et al., 2021), with a scaling factor $\alpha = 32$, rank $r = 8$, and a dropout rate of 0.1. Unless otherwise noted in our test-time scaling analysis (Section 3.2), we set the default depth exit threshold $\lambda_{\text{depth}}$ to 0.6 and the decoding step count $\lambda_{\text{step}}$ to 2. We set the loss coefficient for the language-to-latent alignment (Eq. 9) to $\alpha = 1.0$, and similarly, the loss coefficient for shortcut learning (Eq. 13) to $\beta = 1.0$. Fine-tuning is conducted for 8 epochs using the AdamW optimizer (Loshchilov & Hutter, 2018), with a maximum learning rate of $2 \times 10^{-5}$, $\beta_1 = 0.9$, $\beta_2 = 0.99$, and a warmup over 6% of total training steps. We use a batch size of 2. Training is performed on a single NVIDIA RTX A5000 (24 GB) GPU, requiring approximately 26 hours for LLaMA 3.2 1B and 5 hours for GPT-2 (124M) for an 8-epoch run. All experiments are conducted over four independent runs with different random seeds. We report the average results across these runs to ensure statistical stability.

