# OpenReview forum: "System-1.5 Reasoning: Traversal in Language and Latent Spaces with Dynamic Shortcuts"
_NeurIPS.cc/2025/Conference — NeurIPS 2025 poster_

### Official Review · Reviewer_hpM3 · 2025-06-20

**Clarity:** 1
**Significance:** 3
**Originality:** 3
**Rating:** 4
**Confidence:** 2

**Summary:**

This paper introduces System-1.5 Reasoning, a novel adaptive reasoning framework designed to optimize computational efficiency by combining latent-space reasoning and early-exit mechanisms. The framework utilizes two shortcuts, depth shortcut (DS) and step shortcut (SS), and trained through a two-stage distillation process to selectively allocate computational resources based on step criticality. Empirical evaluations on GSM8K, GSM-HARD, and StrategyQA show that System-1.5 significantly reduces computational overhead and inference time while achieving performance comparable to traditional CoT fine-tuning.

**Questions:**

- The reasoning pattern in Figure 1 is unclear, particularly regarding the process of transformer block computation. Given that computation at a transformer block requires hidden states from all previous steps from the same layer, the figure suggests certain layers are entirely skipped. Is the lightweight LoRA branch on those also skipped? How exactly is the transformer block, especially attention, computed in cases where layers or branches appear entirely skipped?
- Following from the first question, if transformer block computation necessitates some computation for each layer, how does the proposed layer skipping in Equation 1 work during inference? Specifically, is skipping (using a lightweight branch) at an earlier layer independent, or does it influence the decision-making for skipping subsequent layers?
- What is the motivation behind combining depth and step shortcuts in Equation 5 during training, given that they operate as independent branches during inference?
-  Why does Equation 6 focus specifically on optimizing hidden space consistency only for the answer part? Would it not be more logical to optimize either exclusively the reasoning trace or a combined approach?
- Why is joint updating of reasoning steps and answers necessary for the teacher model in Equation 7? How does this joint updating specifically contribute to the model's reasoning capabilities?
- The rationale behind minimizing MSE for critical steps in early layers and minimizing non-critical steps with later layers in Equation 10 is unclear.
- While the paper states that models are built on LLaMA 3.2 1B and GPT-2 124M backbones, it is not clearly specified which backbone is employed in the experimental results reported. Additionally, it would be helpful to know precisely how the baselines are configured with respect to these backbones to ensure fair comparison.

**Ethical Concerns:**

["NO or VERY MINOR ethics concerns only"]

**Final Justification:**

This is a technically solid work. Through the rebuttal, I understood some of the approaches I previously misunderstood in my review. As the authors also present strong experimental results, I am raising my score to 4.

My main concerns remain the clarity of the writing and the limited scale of experimentation. I believe these issues can be better addressed in a later revision.

**Limitations:**

Please see Questions and Weaknesses above.

**Paper Formatting Concerns:**

No formatting issue

**Quality:**

3

**Strengths And Weaknesses:**

**Strength:**
- The paper effectively combines latent-space reasoning with early-exit mechanisms, proposing a well-motivated approach to reduce computational cost in reasoning.
- Empirical results on benchmarks such as GSM8K, GSM-HARD, and StrategyQA consistently demonstrate that System-1.5 Reasoning maintains high performance while substantially accelerating inference.
- Ablation study validates the effectiveness of the proposed architecture and highlights its flexibility in controlling computational budget at inference time.

**Weakness:**
- Several technical details regarding the computation process of each layer and the exact implementation of shortcuts are unclear (see Questions).
- Motivation behind certain algorithmic choices and module designs remains insufficiently justified (see Questions).
- The author does not clearly illustrate or examine the impact of the early-exit loss in Equations 10 and 11 on the labeled intermediate reasoning steps.
- The evaluation methodology has limitations; specifically, using GSM-HARD as an OOD benchmark is questionable given its close relationship with the GSM8K training dataset. Additional evaluation on other datasets, such as MATH, would enhance the reliability of OOD generalization claims.

---

> ### Author Rebuttal · Authors · 2025-07-31
>
> >  **R5-W1** and **R5-Q1**: Clarify how transformer block computation and attention are handled
>
> **R5-A1**: We would like to highlight that the computation flow of Transformer blocks in System-1.5 differs between training and inference: *training* uses a **soft mixture** of shortcut and full paths, while *inference* applies **hard gating** for deterministic execution.
>
> **During training, no computation is skipped**. As described in Eq. (5), the hidden state at each layer $l$ and decoding step $t$ is computed by combining three paths, the standard Transformer path, the depth shortcut from the previous layer, and the step shortcut from the previous step, weighted by a router-predicted gate. This means all layers, including the Transformer block and the LoRA adapter, are computed in every step. Attention computation proceeds as in a standard Transformer, since full hidden states are available for all layers and positions.
>
> **During inference, the router enables early exits based on a threshold**. When a token exits early, all computation beyond the exit layer is skipped, including both the Transformer and LoRA branches. No further attention is computed for that token at deeper layers.
>
> A key challenge in autoregressive decoding is handling attention when tokens exit at different depths, leading to the **missing hidden state** problem: if one token exits early while others continue deeper, the early-exited token lacks the key/value pairs required for attention in the deeper layers. This challenge has been widely studied in prior early-exit models such as LITE and LayerSkip. To address it, System-1.5 adopts the standard solution from these works, copying the hidden state of early-exited tokens forward to the deeper layers. This ensures no disruption in attention computation and preserves compatibility with mainstream inference systems such as HuggingFace Transformers and vLLM.
>
>
> > **R5-W1** and **R5-Q2**: Clarify how layer skipping works during inference.
>
> **R5-A2**: We clarify that Eq. (1) describes training-time behavior, where all branches (standard Transformer, depth shortcut, and step shortcut) are evaluated and combined via a **soft mixture**, weighted by a learned gate. No actual skipping occurs at this stage; instead, the model learns how to weigh different paths. During inference, the model switches to a **hard gating** mechanism.
>
> The rationale is that the router's output weight, learned during training, serves as a confidence score during inference. At each layer, this score is compared against a predefined threshold. If it exceeds the threshold, the token follows the shortcut path and exits early, skipping all subsequent computation, including both the standard Transformer block and the LoRA adapter. Consequently, once a token exits at a particular layer within the current decoding step, it no longer proceeds to deeper layers at the current decoding step, and no further computation is applied. This means that early-exit decisions are final and do not influence or depend on subsequent layers. The resulting hidden state is then forwarded via the step shortcut to initiate the next decoding step.
>
>
> > **R5-W2** and **R5-Q3**: Why are depth and step shortcuts combined?
>
> **R5-A3**: The motivation for combining depth and step shortcuts during training lies in the fact that **they are not entirely independent**.
>
> Specifically, the step shortcut (SS) relies on the hidden states produced by the depth shortcut (DS). SS reuses the intermediate hidden state from the current layer’s early exit (via DS) at the previous decoding step, rather than reinitializing from the first layer as in standard Transformers. Without DS, SS would always operate on final-layer outputs, losing its capacity to capture intermediate reasoning dynamics and effectively collapsing into a trivial skip connection.
>
> To support this interaction, we use a unified router to jointly optimize both branches via Eq. (5), enabling dynamic allocation across both depth and step dimensions, improving flexibility at inference.
>
>
> > **R5-W2** and **R5-Q4**: Why apply consistency only on the answer.
>
> **R5-A4**: Applying consistency loss only to the answer portion is **both intentional and necessary** due to the fundamental nature of latent-space reasoning.
>
> Eq. (6) distills a natural language CoT-finetuned teacher into a latent-space student model. However, latent reasoning operates fundamentally differently from language-space reasoning. In latent reasoning setting, the last-layer hidden states are not used to predict tokens at each intermediate step; instead, they are recursively passed forward as input embedding for subsequent reasoning.
>
> While language-space reasoning **generates one token per decoding step**, latent reasoning represents each step as a continuous thought vector that may **encode multiple tokens or partial reasoning states**. Prior work such as Coconut has demonstrated that these latent representations often exhibit a "superposition" property, encoding multiple intermediate paths simultaneously, akin to a breadth-first search (BFS) in the solution space. This capability is central to the expressive power of latent reasoning.
>
> If we were to enforce consistency across the entire reasoning trace, we would force the latent-space model to mimic the discrete token-level outputs of the teacher. **Since feeding identical last-layer hidden states into the LM head would produce the same token prediction**, this constraint would collapse the latent reasoning process into a token-by-token imitation, thereby eliminating its flexibility and compositionality.
>
> Therefore, we apply the consistency loss only at the final answer step, where both the teacher and student decode into language space. This design anchors the student’s output to the correct answer, while preserving the expressive latent dynamics during intermediate reasoning.
>
>
> > **R5-W2** and **R5-Q5**: Why joint optimization of reasoning and answer in teacher?
>
> **R5-A5**: Eq. (7) represents a standard supervised fine-tuning objective for training a reasoning model, where the model learns to generate both the reasoning steps and the final answer conditioned on the input question. This objective corresponds to the initial training stage in Coconut, and is used in our framework to optimize the natural language CoT teacher model, which captures the full conditional distribution over the reasoning trace followed by the answer.
>
> Jointly optimizing both the reasoning and the answer encourages the teacher to generate coherent and logically structured reasoning paths that directly support the final answer. This structured generation is crucial for providing rich, step-by-step supervision to the latent-space student model, to learn compact and effective latent reasoning representations.
>
>
> > **R5-W3** and  **R5-Q6**: The rationale behind early-exit loss.
>
> **R5-A6**: The rationale lies in the design of the weighting term $e_{l,t}$ in Eq. (10), which **assigns different loss weights** based on the criticality of each reasoning step to encourage the router to appropriately allocate computation.
>
> For *non-critical* steps ($c_t = 0$), the **loss weight increases** with layer depth, as defined by $e_{l,t} = \frac{\sum_{i=1}^l i}{\sum_{j=1}^L j} = \frac{l(l+1)}{L(L+1)}$. This means shallow layers are assigned lower weights (losses) and deeper layers higher weights (losses), encouraging non-critical steps to exit earlier and avoid unnecessary computation.
>
> Conversely, for *critical* steps ($c_t = 1$), the **loss weight decreases** with layer depth, given by $e_{l,t} = 1 - \frac{\sum_{i=1}^l i}{\sum_{j=1}^L j} = 1 - \frac{l(l+1)}{L(L+1)}$. This penalizes early exits at shallow layers more heavily, encouraging the model to delay exiting and allocate more computation for these important steps.
>
> > **R5-Q7**: Clarification on backbone models.
>
> **R5-A7**: Thank you for the helpful suggestion. We clarify that the results on GSM8K and StrategyQA are based on GPT-2 124M, following the setup used in Coconut. Specifically, we use the GPT2LMHeadModel from HuggingFace with the following architecture: $n_{\text{emb}} = 768$, $n_{\text{layer}} = 12$, $n_{\text{head}} = 12$. The model is trained in float32 precision. Since GPT-2 was pretrained without a padding token, we set the pad_token_id to match the eos_token_id, as done in Coconut.
>
> For GSM-HARD, we use LLaMA 3.2B, consistent with the LayerSkip baseline. The model is obtained from the official HuggingFace release, and we train System-1.5 using the same configuration: $n_{\text{emb}} = 2048$, $n_{\text{layer}} = 16$, $n_{\text{head}} = 32$. Training is conducted in bfloat16 precision, as specified by LayerSkip.
>
> > **R5-W4**: Additional evaluation on MATH dataset.
>
> **R5-A8**: Thank you for the thoughtful suggestion. To better assess out-of-distribution generalization, we conducted additional evaluation on the MATH dataset using the LLaMA 3.2 1B backbone models. The results are shown below:
>
> | Method      | Acc. (%)  | # Steps | FLOPs r. | Speedup    |
> | ----------- | --------- | ------- | -------- | ---------- |
> | CoT         | 30.12     | 26      | –        | –          |
> | LayerSkip   | 27.92     | 32      | 1.77×    | 1.45×      |
> | iCoT        | 24.18     | 2       | 1.02×    | 13.45×     |
> | Coconut     | 25.31     | 2       | 1.02×    | 11.98×     |
> | CODI        | 27.04     | 2       | 1.02×    | 13.37×     |
> | Pause Token | 27.76     | 38      | 1.00×    | 0.80×      |
> | System-1.5  | 29.85 | 2       | 1.95×    | **20.27×** |
>
> while all models exhibit a performance drop on the MATH dataset, owing to its increased complexity in algebra and calculus compared to GSM8K, System-1.5 still achieves accuracy nearly matching its CoT teacher (29.85% vs. 30.12%), while significantly outperforming all baselines in inference efficiency. These results highlight the strong generalization of System-1.5, even on more challenging out-of-distribution reasoning tasks.

---

> > ### Comment · Reviewer_hpM3 · 2025-08-01
> >
> > Thank you for your responses, I understood some of the approaches I previously misunderstood in my review. As the authors also present strong experimental results, I am raising my score to 4.
> >
> > My main concerns remain the clarity of the writing and the limited scale of experimentation. I believe these issues can be better addressed in a later revision.

---

> > > ### Author Response · Authors · 2025-08-01
> > > **Thanks for the score update and suggestions**
> > >
> > > Thank you for your prompt response and for raising your score. We appreciate your suggestions regarding clarity and experimental scale, and will continue to improve both in the next revision. Please feel free to reach out with any further questions.

---

### Official Review · Reviewer_zebB · 2025-06-23

**Clarity:** 3
**Significance:** 2
**Originality:** 3
**Rating:** 4
**Confidence:** 4

**Summary:**

This paper mitigates the inefficiency of Chain-of-Thought reasoning in Large Language Models. While standard CoT improves reasoning accuracy (System-2), it is slow. Existing latent-space reasoning methods improve speed by avoiding text generation but are still suboptimal as they apply uniform computation to all reasoning steps, regardless of their complexity.  To overcome this, the authors propose System-1.5 Reasoning, an adaptive framework that dynamically allocates computation via Depth Shortcut and Step Shortcut. Experiments on mathematical (GSM8K, GSM-HARD) and commonsense (StrategyQA) reasoning tasks show that System-1.5 Reasoning achieves accuracy comparable to standard CoT fine-tuning while providing massive efficiency gains, including an over 20x inference speedup on GSM8K.

**Questions:**

The shortcut learning stage is guided by 'reasoning criticality' labels derived from atomic thought decomposition. How sensitive is the final performance of System-1.5 Reasoning to the quality and granularity of these annotations?

**Ethical Concerns:**

["NO or VERY MINOR ethics concerns only"]

**Limitations:**

As mentioned above.

**Quality:**

2

**Strengths And Weaknesses:**

**Strengths**
1. The performance gains reported are impressive. Achieving a >20x speedup and >90% token reduction on benchmarks like GSM8K, while maintaining the accuracy of the much slower CoT baseline.
2. The idea of combining both vertical (depth-wise) and horizontal (step-wise) shortcuts is novel.

**Weaknesses**
1. The shortcut learning stage relies on criticality labels generated by an external "atomic thought decomposition" process. This introduces a dependency and a potential point of failure. How sensitive is the performance of System-1.5 Reasoning to the quality of these annotations? Have the authors considered scenarios with noisy or imperfect criticality labels?
2. The proposed framework is fundamentally supervised, relying entirely on distillation from a teacher model that generates explicit, pre-defined CoT paths. The critical supervisor signal—the reasoning criticality labels—is also derived from these static teacher traces. This makes it unclear how the method could be extended to a Reinforcement Learning (RL) setting like DeepSeek R1.
3. The full framework is quite involved, requiring a pre-trained base model, a CoT-finetuned teacher, a System-2 student, a criticality annotation pipeline, and finally the System-1.5 model.

---

> ### Author Rebuttal · Authors · 2025-07-31
>
> > **R4-Q1** and **R4-Q4**:: Concern about how sensitive System-1.5 is to the quality of criticality annotations used for shortcut learning.
>
> **R4-A1** and **R4-A4**: We thank the reviewer for highlighting this important concern. While the shortcut learning stage relies on criticality labels derived from atom-of-thought decomposition, we find that **System-1.5 Reasoning is relatively robust to label imperfections due to its use of coarser-grained, step-level supervision rather than direct token-level label**. This design mitigates the impact of noise and helps bridge the reasoning pattern gap between language space and latent space.
>
> Specifically, the goal of shortcut learning is to allocate computation adaptively in latent space according to reasoning difficulty inferred from language supervision. However, latent-space reasoning differs fundamentally from language-space reasoning: in language space, **a decoding step corresponds to a discrete token**, whereas in latent space, **a step is a continuous thought vector that may encode multiple tokens or reasoning states**. Prior work such as Coconut has shown that these latent vectors often exhibit "superposition", representing parallel paths or intermediate states akin to breadth-first search (BFS).
>
> This structural difference makes token-level supervision too fine-grained and misaligned with the nature of latent steps. Therefore, we adopt sub-step–level annotations to provide supervision at a more appropriate granularity. Using atom-of-thought decomposition, we convert multi-step reasoning traces into a Directed Acyclic Graph (DAG), where each node corresponds to a minimal reasoning unit. Nodes with logical dependencies are labeled as critical, and independent nodes as non-critical. This labeling is then propagated to all tokens within each sub-step.
>
> To directly evaluate the sensitivity to annotation granularity and quality, we compare our step-level labeling against a fine-grained token-level supervision strategy, where criticality is estimated from the teacher model’s token-wise confidence scores, low-confidence tokens (confidence < 0.5) are treated as critical and others as non-critical. The results are shown below:
>
> | Labeling Strategy   | Accuracy (GSM8K) | Speedup    | Notes                           |
> | ------------------- | ---------------- | ---------- | ------------------------------- |
> | Coconut             | 36.75            | 11.98×     | No shortcut supervision         |
> | Token-level (Conf.) | 35.94            | 20.25×     | Token-wise difficulty heuristic |
> | Step-level (Ours)   | **46.66**        | **20.27×** | Based on atom-of-thought DAG    |
>
> As shown, token-level supervision not only underperforms our step-level method but also falls below the Coconut baseline. This highlights that fine-grained supervision can introduce instability and noise that harms both performance and efficiency. In contrast, our step-level approach offers structurally coherent and stable signals that better guide shortcut optimization in latent space.
>
>
> > **R4-Q2**: Concern about extending System-1.5 to RL settings without ground-truth CoT.
>
> **R4-A2**: We thank the reviewer for this insightful and forward-looking question. While System-1.5 Reasoning currently follows a supervised fine-tuning paradigm, we agree that extending it to a reinforcement learning (RL) setting, particularly in the absence of explicit ground-truth CoT traces, is both valuable and non-trivial. We are actively exploring this direction by **reframing the self-distillation process into a self-reward paradigm**.
>
> Specifically, at a high level, System-1.5 is trained in two stages:
> (1) distilling language-based CoT into latent-space continuous thoughts (language-to-latent alignment), and
> (2) distilling full-path latent reasoning into adaptive shortcut policies (shortcut learning).
>
> A potential RL-compatible formulation is to reinterpret the first stage as reward modeling, where the goal is to learn a reward function that captures the quality of latent reasoning trajectories, for example, their ability to reach correct answers with minimal computation. This learned reward model could then be used to guide the router policy through reinforcement learning, effectively enabling shortcut selection without explicit CoT annotations.
>
> In this formulation, the main challenge lies in reward model training: acquiring high-quality signals that reflect step-wise reasoning value. Here, teacher models could still serve as a source of weak supervision, providing approximate scoring based on trajectory alignment or final output correctness. We view this as a promising direction for extending System-1.5 beyond supervised learning, and will include this discussion in the revised version to clarify the RL-compatible potential of our framework.
>
>
> > **R4-Q3**: Concern about the complexity of the full framework
>
> **R4-A3**: Thank you for the thoughtful question. While System-1.5 Reasoning adopts a two-stage training procedure, we emphasize that its overall complexity remains practical for both research and deployment. This is due to two key factors: (1) the **two stages are modular and loosely coupled**, and (2) the **overall training efficiency is competitive**.
>
> First, the System-2 student model is compatible with other latent reasoning architectures and can be swapped out without retraining the full pipeline. As shown in Section 3.2, we successfully replaced our student model with alternative latent reasoning backbones such as Coconut and CODI, while maintaining strong performance. This modularity allows the framework to generalize across reasoning strategies.
>
> Second, the training process is efficient in practice. In Stage 1 (language-to-latent alignment), we adopt a distillation-based approach with **teacher forcing**, extracting the last-layer hidden states from a CoT-trained teacher model and using stop-gradient to guide the student model. This enables parallel training of the Transformer backbone without requiring step-wise autoregressive decoding or curriculum scheduling, as used in prior methods like Coconut or iCoT. As a result, this stage achieves significantly higher throughput.
>
> In Stage 2 (shortcut learning), only **lightweight router and adapter** modules are trained, while the base Transformer remains frozen. These adapters introduce fewer than 1% trainable parameters and follow the efficiency principles of LoRA-style tuning, making this phase both fast and hardware-friendly.
>
> Following your suggestion, we provide a summary of training cost and inference speedup using GPT-2 as the base model:
>
> | Method               | Training Time (hrs) | # Trainable Params | Inference Speedup |
> | -------------------- | ------------------- | ------------------ | ----------------- |
> | CoT Fine-tuning      | 4.2                 | 100%               | 1.00×             |
> | Coconut              | 8.6                 | 100%               | 11.98×            |
> | System-1.5 (Stage 1) | 4.3                 | 100%               | –                 |
> | System-1.5 (Stage 2) | 0.6                 | \~1%               | –                 |
> | System-1.5 (Overall) | 4.9 (4.3 + 0.6)     | –                  | **20.27×**        |
>
> These results show that System-1.5 requires only 4.9 hours to train, nearly matching CoT fine-tuning (4.2 hrs) and significantly outperforming Coconut in both training time and inference efficiency. Despite its two-stage structure, the second stage updates only ~1% of parameters, keeping overhead minimal. Most notably, System-1.5 achieves a 20.27× inference speedup, confirming its practical advantages for scalable deployment.

---

> > ### Comment · Reviewer_zebB · 2025-08-06
> >
> > Thank you for the detailed reply. I'm willing to keep my positive score. Good luck.

---

### Official Review · Reviewer_zDgS · 2025-06-24

**Clarity:** 3
**Significance:** 4
**Originality:** 4
**Rating:** 4
**Confidence:** 3

**Summary:**

This paper carries out an important and innovative study to decrease the inference cost of reasoning LLMs by training LLM with the mechanisms of depth and step shortcuts. By varying the inference cost on each token by weighing their significance, the model focuses on more important reasoning token and step, and skip the trivial tokens early.

Overall, the proposed method is novel and the writing is clear. However, I have questions on the construction of the dataset for shortcut learning. The authors do not specifically illustrate the rationality of this process and the trustworthiness of the method. Meanwhile, I am curious why the sub-step criticality can be used to determine the criticality of the token in this step?

**Questions:**

As listed above,

**Ethical Concerns:**

["NO or VERY MINOR ethics concerns only"]

**Final Justification:**

I think this paper is novel and solid. I recommend acceptance in this case.

**Limitations:**

In appendix. Need to be moved to main content.

**Paper Formatting Concerns:**

No.

**Quality:**

3

**Strengths And Weaknesses:**

I think this paper has many strengths. Hence, I will not list the strengths specifically since I think the paper can be accepted if the questions on dataset construction can be answered,

The main weakness is: how the dataset for shortcut learning is constructed, Why atom-of-thought to construct directed acyclic graph is one decent way? Is there any analysis or methods to verify the correctness? Meanwhile, why the importance assessment on the sub-step can be used to assess the token importance?

---

> ### Author Rebuttal · Authors · 2025-07-31
>
> We sincerely thank the reviewer for their encouraging comments and thoughtful questions, and we appreciate the opportunity to clarify the construction process of the shortcut learning dataset and elaborate on the motivations and insights behind its design.
>
> > **R3-Q1**: Why is atom-of-thought and DAG construction a justified strategy? and
>
> > **R3-Q2**: How is sub-step importance meaningfully mapped to token-level importance?
>
> **R3-A1** and **R3-A2**: The reason we adopt atom-of-thought and DAG construction is that **we indeed require higher-level criticality annotation beyond token-wise labels**, in order to bridge the reasoning pattern gap between language-space and latent-space.
>
> Specifically, the goal of shortcut learning is to adaptively allocate computation in latent space based on reasoning difficulty estimated from language-space supervision. However, latent-space reasoning operates fundamentally differently from language-space reasoning: in language space, **each step corresponds to a discrete token**, whereas in latent space, **each step represents a continuous thought vector that may encode multiple tokens** or partial reasoning states. Prior work such as Coconut has empirically and theoretically shown that these latent representations often exhibit a "superposition" property, representing multiple potential paths or intermediate states simultaneously, akin to breadth-first search (BFS) in the solution space.
>
> This structural mismatch makes token-level supervision too fine-grained and misaligned with the latent reasoning pattern. Instead, we adopt sub-step–level supervision that better reflects the hierarchical nature of multi-step reasoning. To construct these sub-steps, we apply atom-of-thought decomposition, which organizes a chain-of-thought sequence into a Directed Acyclic Graph (DAG), where each node corresponds to a minimal, self-contained reasoning unit. Nodes that depend on others (i.e., derived steps) are labeled as critical, while logically independent or restatement-type nodes are labeled as non-critical.
>
> This labeling offers a structured and interpretable form of reasoning supervision. During shortcut learning, we assign each token the criticality label of its enclosing sub-step, resulting in segment-wise supervision: all tokens in a critical sub-step are labeled as critical, while those in non-critical sub-steps are not. **This design naturally allows the same token to receive different labels depending on its functional role in different reasoning contexts**. Importantly, since shortcut learning operates entirely on hidden states, rather than through the LM head or vocabulary output, this label propagation does not introduce bias tied to token identity and remains fully compatible with latent-space modeling.
>
> To empirically validate this decision, we compare our step-level supervision with a fine-grained token-level labeling strategy based on **teacher model confidence**, where low-confidence tokens (confidence < 0.5) are treated as critical and others as non-critical. The results are summarized below:
>
> | Labeling Strategy   | Accuracy (GSM8K) | Speedup    | Notes                           |
> | ------------------- | ---------------- | ---------- | ------------------------------- |
> | Coconut             | 36.75            | 11.98×     | No shortcut supervision         |
> | Token-level (Conf.) | 35.94            | 20.25×     | Token-wise difficulty heuristic |
> | Step-level (Ours)   | **46.66**        | **20.27×** | Based on atom-of-thought DAG    |
>
> As shown, the token-level labeling strategy leads to significantly weaker performance, even underperforming the Coconut baseline. A possible reason would be the noise and instability inherent in token-wise difficulty estimation, which lacks the structural coherence needed for guiding shortcut optimization in latent space. In contrast, our step-level supervision, derived from atom-of-thought decomposition, provides more stable and meaningful guidance for adaptive reasoning.
> We will incorporate these clarifications on design motivations and the comparative analysis in the revised version to further support the trustworthiness and rationale behind our shortcut learning supervision.

---

### Official Review · Reviewer_ws8G · 2025-07-03

**Clarity:** 2
**Significance:** 3
**Originality:** 3
**Rating:** 4
**Confidence:** 4

**Summary:**

This paper proposes an approach to achieve adaptive reasoning to improve the efficiency of reasoning via language tokens. The approach is evaluated on two categories of tasks (GSM8K and StrategyQA) to demonstrate the it achieves the same accuracy as language CoT but significantly improves inference efficiency. The main approach is distillation from language CoT with a curriculum.

**Questions:**

1. What's the comparison with different baselines in terms of training throughput (speed)?
2. The approach is evaluated in both math reasoning and commonsense reasoning. But how does the performance differ with problems of different levels of difficulty?
3. Regarding evaluating controllable test-time scaling, can you also report pass@k with increasing k and compare it to the trend of baselines?

**Ethical Concerns:**

["NO or VERY MINOR ethics concerns only"]

**Final Justification:**

The authors provided further results which have addressed some of the limitations in my original review.

**Limitations:**

yes

**Paper Formatting Concerns:**

no concern

**Quality:**

2

**Strengths And Weaknesses:**

Strength:
The paper addresses an important problem of improve reasoning efficiency and provides new empirical results on reasoning in latent space. The design of the method to be adaptive and yet controllable is an appealing property for test-time compute scaling.

Weaknesses:
The training is very complex and lacks in-depth ablations to understand the importance of different components, e.g. the importance of curriculum, what are important factors for optimization, e.g. should the router be jointly learnt, how does freezing parameters affect training, etc.

---

> ### Author Rebuttal · Authors · 2025-07-31
>
> > **R2-Q1**: Missing ablations on curriculum and optimization decisions.
>
> **R2-A1**: We would like to clarify that **the original submission includes the ablations the reviewer mentions**, and we appreciate the opportunity to further highlight them here.
>
> First, regarding the role of curriculum, we **compare curriculum-based language-to-latent alignment (used in Coconut) with our direct distillation strategy** in the first paragraph of Section 3.2. The results show that curriculum learning leads to suboptimal performance.
>
> Second, in the second paragraph of Section 3.2, we **analyze optimization strategies for training the router**. Specifically, we compare joint learning, where both Transformer and router parameters are optimized together, with our proposed two-stage approach. The results show a significant performance drop under joint learning, likely due to optimization conflicts between the two parameter groups.
>
> Third, to further analyze component-wise contributions, we **conduct additional ablations isolating the depth shortcut (DS) and step shortcut (SS)**. In the DS-only setting, we disable SS and feed early-exited hidden states back to the first layer in the next decoding step, enabling adaptive vertical reasoning without horizontal reuse. In the SS-only setting, we disable early exits but retain the final-layer routing, allowing last-layer features to be reused in the next decoding step, simulating horizontal step skipping without dynamic depth traversal.
>
> | Method     | Acc. (%)  | # Steps | FLOPs r. | Speedup    |
> | ---------- | --------- | ------- | -------- | ---------- |
> | Coconut    | 36.75     | 2       | 1.02×    | 11.98×     |
> | SS-only    | 38.24     | 2       | 0.74×    | 15.26×     |
> | DS-only    | 40.02     | 2       | 0.91×    | 12.53×     |
> | System-1.5 | **46.66** | 2       | 1.95×    | **20.27×** |
>
> We find that both ablated variants still outperform standard latent reasoning baselines like Coconut in efficiency and maintain competitive accuracy, consistent with prior work on language-space early-exit. However, both underperform the full System-1.5 model. Notably, DS-only yields higher accuracy than SS-only. A possible explanation is that DS-only, by enabling routing to both first and last layers, can simulate both full-path and skip-step behavior. However, this does not imply that SS is redundant. In fact, DS-only demonstrates weaker efficiency than SS-only, highlighting that SS plays a crucial role in reducing unnecessary reasoning steps. Taken together, these results underscore the complementary roles of DS and SS. Their integration enables System-1.5 to dynamically allocate computation across both depth and step dimensions, faithfully reflecting the core design of our method.
>
>
>
> > **R2-Q2**: Missing comparison of training speed
>
> **R2-A2**: Thank you for the question. We would first like to clarify that although System-1.5 Reasoning adopts a two-stage training procedure, **its overall training efficiency remains competitive**.
>
> In Stage 1 (language-to-latent alignment), our approach leverages a distillation-based setup with **teacher forcing**. Unlike prior latent reasoning methods such as Coconut and iCoT, which rely on step-wise autoregressive decoding and curriculum scheduling, System-1.5 extracts last-layer hidden states from the CoT-trained teacher model, applies stop-gradient, and uses these states as ground-truth features for the student model. This setup enables parallelized Transformer training without autoregressive rollout, significantly improving throughput.
>
> In Stage 2 (shortcut learning), only the **lightweight router-adapter** modules are updated, with the base Transformer parameters frozen. These adapter modules require gradients for less than 1% of model parameters, making this phase highly efficient and comparable to standard LoRA-style tuning.
>
> We report the end-to-end training time and inference efficiency of System-1.5 in comparison to CoT and Coconut baselines using GPT-2 (124M) on the same device.
>
> | Method               | Training Time (hrs) | # Trainable Params | Inference Speedup |
> | -------------------- | ------------------- | ------------------ | ----------------- |
> | CoT Fine-tuning      | 4.2                 | 100%               | 1.00×             |
> | Coconut              | 8.6                 | 100%               | 11.98×            |
> | System-1.5 (Stage 1) | 4.3                 | 100%               | –                 |
> | System-1.5 (Stage 2) | 0.6                 | \~1%               | –                 |
> | System-1.5 (Overall) | 4.9 (4.3 + 0.6)     | –                  | **20.27×**        |
>
> From the results, System-1.5 requires only 4.9 hours to train, slightly more than CoT (4.2 hrs) but significantly less than Coconut (8.6 hrs). Notably, its second-stage shortcut learning phase updates only ~1% of the model parameters, making it lightweight and efficient to fine-tune.
>
>
> > **R2-Q3**: No analysis on how performance varies with problem difficulty.
>
> **R2-A3**: We appreciate the question and note that although GSM8K and StrategyQA lack explicit difficulty annotations, we analyze performance variation by comparing GSM8K to the more challenging GSM-HARD, which reveals **consistent degradation across all methods and reduced efficiency gains**.
>
> Specifically, GSM-HARD increases difficulty by modifying GSM8K problems and serves as an out-of-domain evaluation, compounding the challenge. As shown in Table 1, performance drops for all methods, CoT (46.94%  $\rightarrow$ 38.32%), Coconut (36.75% $\rightarrow$  28.25%), and System-1.5 (46.66% $\rightarrow$  38.28%), confirming increased difficulty. Despite this, System-1.5 maintains accuracy comparable to CoT and outperforms latent reasoning baselines like Coconut and iCoT in the harder setting. Inference speedups are also lower on GSM-HARD, suggesting that increased reasoning complexity limits opportunities for shortcut execution, aligning with the intuition that harder problems reduce the potential for early exits or step skipping.
>
>
> > **R2-Q4**: Missing pass@k analysis under test-time scaling.
>
> **R2-A4**: Thank you for the question. We would first like to clarify that **applying standard pass@k evaluation is not straightforward and may be inappropriate for latent-space reasoning methods** such as System-1.5 Reasoning, Coconut, and iCoT.
>
> These methods differ fundamentally from language-space reasoning CoT: they do not decode intermediate steps into natural language. Instead, they operate entirely in latent space by passing internal hidden states, either final-layer representations or early-exit features (in the case of System-1.5), as input embeddings to the next step. Since latent steps do not produce vocabulary distributions via the LM head, top-k sampling or beam search cannot be applied to generate diverse reasoning paths during intermediate steps.
>
> As a result, pass@k evaluation for latent methods can only be reasonably applied to the final answer decoding step. While one could vary the depth exit threshold to simulate different paths, this would alter the reasoning behavior itself and introduce confounding factors, making it an unfair basis for comparison.
>
> Nevertheless, for completeness, we report pass@k on the final answer token (only) using top-k sampling at inference for latent reasoning baselines (System-1.5, Coconut), and apply standard top-k sampling across all tokens for CoT baselines:
>
> | Method                                 | Pass@1 | Pass@5 | Pass@10 | Pass@100 |
> |----------------------------------------|-------:|-------:|--------:|---------:|
> | CoT                                    | 46.94  | **48.71**  | **51.23**   | **51.92**    |
> | Coconut (step = 2)                     | 36.75  | 37.88  | 38.15   | 38.60    |
> | Coconut (step = 4)                     | 37.03  | 38.02  | 38.47   | 39.01    |
> | Coconut (step = 8)                     | 37.22  | 38.31  | 38.69   | 39.12    |
> | Coconut (step = 32)                    | 37.44  | 38.59  | 39.03   | 39.55    |
> | System-1.5 (depth = 0.6, step = 2)     | 46.66  | 48.03  | 48.44   | 48.81    |
> | System-1.5 (depth = 0.6, step = 4)     | 47.08  | 48.22  | 48.66   | 48.93    |
> | System-1.5 (depth = 0.6, step = 8)     | 47.58  | 48.36  | 48.80   | 49.12    |
> | System-1.5 (depth = 0.6, step = 32)    | **48.67**  | 48.48  | 48.92   | 49.27    |
>
>
> From these results, CoT shows clear gains ($\sim$5%) with increasing $k$ (1, 5, and 10), benefiting from diversity across multiple sampled reasoning chains. In contrast, System-1.5 and Coconut exhibit much smaller improvements (typically <1%), as top-$k$ sampling is only applied at the final answer token, while all intermediate reasoning steps remain deterministic. This highlights a structural property of latent-space methods: increasing $k$ primarily affects the surface-level output, rather than altering the internal reasoning trajectory.
>
> Finally, we note that large-$k$ sampling (e.g., $k = 100$) in CoT can be computationally expensive and yields diminishing returns. For instance, pass@10 improves from 51.23% to only 51.92% at pass@100, suggesting performance saturation. This indicates that, even with significantly increased sampling budget, language-space reasoning has limited ability to improve its correctness beyond a certain point.
>
> In contrast, System-1.5 remains both effective and efficient without depending on high-$k$ sampling. Notably, the configuration with depth = 0.6 and step = 32 achieves a reasoning length and computational cost comparable to pass@1 CoT, while matching CoT’s pass@5 accuracy (48.67% vs. 48.71%). This demonstrates that System-1.5’s latent, shortcut-based reasoning offers a more robust and structurally efficient alternative to traditional CoT sampling-based approaches.

---

### Official Review · Reviewer_YTmj · 2025-07-03

**Clarity:** 2
**Significance:** 3
**Originality:** 3
**Rating:** 4
**Confidence:** 4

**Summary:**

This paper introduces System-1.5 Reasoning, an adaptive reasoning framework for large language models that uses dynamic shortcuts in latent space to balance efficiency and performance in chain-of-thought reasoning. The approach leverages two shortcut types—model depth shortcuts and decoding step shortcuts—allowing the system to allocate more computation to important reasoning steps and less to easier ones. Experiments on standard reasoning benchmarks show that System-1.5 Reasoning matches or surpasses existing baselines, while significantly reducing inference cost.

**Questions:**

- How is the routing for depth and step shortcuts initialized or warmed up? Does the router start with random predictions, and are there training curves available showing its learning progress?
- Is this method compatible with mainstream inference engines (such as vllm or sglang)? These engines are already highly optimized for fast inference. If System-1.5 is not compatible, then its practical advantage may be limited despite the speed improvements shown in the paper.

**Ethical Concerns:**

["NO or VERY MINOR ethics concerns only"]

**Final Justification:**

The author has solved my problem, and I have increased the score to 4.

**Limitations:**

yes

**Quality:**

2

**Strengths And Weaknesses:**

**Strengths**
- The proposed framework is well-motivated, introducing flexible, dynamic computational shortcuts at both the model depth and decoding step levels.
- The method delivers substantial speed-ups—up to 20× faster inference on GSM8K—while maintaining or even improving accuracy over prior work.

**Weaknesses**
- The experimental section feels a bit thin. Table 1 only reports final results and lacks detailed ablation studies. For example, how do depth and step shortcuts individually contribute to speed and accuracy? Figure 3 examines student selection and joint/full-parameter learning, but does not directly analyze the role or interaction between depth and step shortcuts. What happens if only one type of shortcut is used?
- There are no concrete case studies showing how depth and step shortcuts are routed in practice. How does the system decide which shortcut to take for different types of questions? Are there potential issues with load balancing?
- All experiments are conducted on relatively small models (GPT-2 124M, LLaMA 3.2B). While understandable for research, this limits the relevance for real-world applications, where much larger models (>7B) are common.

---

> ### Author Rebuttal · Authors · 2025-07-31
>
> We thank the reviewer for the constructive feedback and for recognizing the strong motivation behind our framework, as well as its substantial speed-ups while maintaining performance.
>
> > **R1-Q1**: Individual contributions and interaction of depth vs. step shortcuts not analyzed.
>
> **R1-A1**: We would first clarify that the **depth shortcut (DS) and step shortcut (SS) are not fully independent**, making it challenging to isolate their effects directly.
>
> Specifically, SS depends on the hidden states produced by DS, i.e., it reuses early-exited hidden states from intermediate layers instead of restarting from the first layer at every step as in standard Transformers. Without DS, SS would always operate on final-layer outputs, removing its capacity for effective intermediate reasoning and degenerating into a trivial skip.
>
> To better address this concern, we introduced architectural and training adjustments to approximate ablations with minimized interdependence. In the DS-only setting, we removed SS and routed early-exited hidden states back to the first layer at the next decoding step, effectively enabling adaptive early exit without horizontal reuse. In the SS-only setting, we disabled early exits but retained routing at the final layer, allowing the model to reuse the last-layer hidden state to skip a decoding step and continue reasoning from the first layer, thus simulating adaptive step skipping without dynamic depth control.
>
> These modifications inevitably reduce the original flexibility and design intent of System-1.5 and the results are summarized as follows:
>
> | Method     | Acc. (%)  | # Steps | FLOPs r. | Speedup    |
> | ---------- | --------- | ------- | -------- | ---------- |
> | Coconut    | 36.75     | 2       | 1.02×    | 11.98×     |
> | SS-only    | 38.24     | 2       | 0.74×    | 15.26×     |
> | DS-only    | 40.02     | 2       | 0.91×    | 12.53×     |
> | System-1.5 | **46.66** | 2       | 1.95×    | **20.27×** |
>
> Both ablation settings still achieve comparable performance and improved efficiency over standard latent reasoning baselines like Coconut, consistent with prior studies on early-exit in language space. However, both underperform the full System-1.5 model. Specifically, we observe that DS-only achieves better accuracy than SS-only. A likely explanation is that DS-only is functionally a superset of SS-only: during inference, choosing either the first-layer or last-layer exit effectively simulates both full-layer traversal and skip-step behavior, as seen in DS-only execution. Nonetheless, this does not imply that SS is redundant. In fact, the DS-only variant demonstrates weaker efficiency compared to SS-only, suggesting that SS plays a crucial role in reducing unnecessary reasoning steps.
>
> Taken together, these results underscore the complementary nature of DS and SS. Their integration allows the model, under a fixed number of reasoning steps, to dynamically select the most efficient and effective path by allocating more computation to hard steps and skipping simpler ones—capturing the intended design of System-1.5 Reasoning.
>
> > **R1-Q2**: Unclear how shortcut routing works in practice and how decisions are made; possible load balancing concerns.
>
> **R1-A2**: We would like to clarify that **shortcut decisions are made adaptively at inference time**, based on step-level reasoning criticality.
>
> As described in Lines 147–155, the depth shortcut (DS), i.e., early exit at intermediate layers, is controlled by a depth exit threshold $\lambda_\text{depth}$. This threshold is applied to the confidence score output from the router module, which is trained with an early-exit loss to correlate layer depth with reasoning difficulty: critical steps are routed to deeper layers, while non-critical ones exit early at shallow layers.
>
> The step shortcut (SS) follows the routing outcome of DS. Specifically, hidden states from early-exited layers are directly passed to the next decoding step within the same layer, instead of restarting from the first layer. This design ensures that the SS mechanism builds on the vertical routing structure provided by DS, forming a coherent and efficient reasoning trajectory.
>
> Regarding potential load balancing concerns, we note that this adaptive shortcut mechanism not only mitigates imbalance but also offers fine-grained control over the reasoning budget during inference. As shown in Section 3.2, by adjusting the depth threshold, users can easily scale computation based on available resources or task complexity. This tunability in latent reasoning space provides a simple yet effective alternative to more complex tree-based exploration strategies or multi-path consistency mechanisms used in language-space test-time scaling.
>
>
> > **R1-Q3**: Small model sizes limit applicability to real-world large-model settings.
>
> **R1-A3**: Thank you for the suggestion. We initially adopted GPT-2 (124M) and LLaMA-3.2-1B as backbones to **ensure fair comparisons with previous latent reasoning baselines** such as Coconut and CODI, which are also built on small models.
>
> To better address your concern, we have scaled System-1.5 Reasoning to two widely used 7B-scale open-source backbones: LLaMA-3.1-8B and Qwen2.5-7B. The results are shown as follows:
>
> | Model          | Method     | Acc. (%) | Speedup |
> | -------------- | ---------- | -------- | ------- |
> | GPT-2 (124M)   | CoT        | 46.94    | –       |
> |                | System-1.5 | 46.66    | 20.27×  |
> | LLaMA-3.1 (8B) | CoT        | 78.42    | –       |
> |                | System-1.5 | 77.85    | 29.36×  |
> | Qwen2.5 (7B)   | CoT        | 76.90    | –       |
> |                | System-1.5 | 75.88    | 29.02×  |
>
>
> We find that System-1.5 continues to benefit from model scaling, showing consistent performance improvements compared to its 1B-scale counterpart. Moreover, the relative speedup over CoT baselines increases with model size. For example, while System-1.5 achieves a speedup of 20x at 1B scale, it reaches 29x at 8B. This implies that larger models introduce more computational redundancy, which can be more effectively reduced through dynamic shortcuts, leading to better efficiency while maintaining or even improving reasoning accuracy.
>
>
> > **R1-Q4**: Unclear how the router is initialized or warmed up; training curve of the router.
>
> **R1-A4**: **The router is initialized with standard techniques** and progressively learns to allocate computation, with **shortcut paths gradually activated from zero initialization**.
>
> Specifically, as defined in Eq. (5), the hidden state at each layer is a weighted combination of the standard Transformer path and the shortcut paths, where the router-predicted weight $w$ (output of a lightweight FFN with sigmoid) controls the balance:
>
> $$
> h_{l,t}^{\text{system-1.5-training}} = \left( g_{l-1}(h_{l-1,t}) + g_l(h_{l,t-1}) \right) \cdot w + f_l(h_{l-1,t}) \cdot (1 - w)
> $$
>
> The router parameters are initialized from a standard normal distribution, and the shortcut adapters $g(\cdot)$ are initialized to zero as LoRA, ensuring that early in training, the model behaves like a standard Transformer and avoids unstable routing. As training proceeds, the model learns to activate shortcut paths based on supervision.
>
> To quantify this warm-up behavior, we track the $L_2$ norm ratio between shortcut and main-path outputs during training:
>
> $$
> \text{Shortcut Ratio} = \frac{\Vert g_{l-1}(h_{l-1,t}) + g_l(h_{l,t-1}) \Vert_2}{\Vert f_l(h_{l-1,t}) \Vert_2}
> $$
>
> As figures can not be supported in rebuttals, we summarize the dynamics numerically by reporting representative values of the shortcut-to-main-path ratio across training epochs:
>
> | Epoch          | 1    | 2    | 4    | 6    | 8    |
> | -------------- | ---- | ---- | ---- | ---- | ---- |
> | Shortcut Ratio | 0.02 | 0.07 | 0.15 | 0.28 | 0.36 |
>
> This consistent growth confirms that routing behavior is learned stably over time, progressively shifting computation from full-path to shortcut reasoning.
>
>
> > **R1-Q5**: Unclear if System-1.5 is compatible with vLLM/sglang or other optimized inference engines.
>
> **R1-A5**: System-1.5 Reasoning is **fully compatible with** optimized inference engines like vLLM and sglang by adopting standard solutions for early-exit decoding and preserving the vanilla Transformer interface.
>
> The early-exit behavior introduced by System-1.5, where tokens may terminate at different depths, poses a known challenge in efficient batched decoding, commonly referred to as the “**missing hidden state**” problem. Specifically, if one token exits early at a shallow layer and a subsequent token in the same batch continues deeper, then the early-exited token lacks the key/value states required for deeper attention layers during autoregressive decoding.
>
> This issue is not unique to our method and has been well studied in early-exit LLMs such as CALM, LITE, and LayerSkip and early-exit inference framework. The widely adopted solution is to copy the shallow-layer hidden states forward into the deeper layers when needed, allowing consistent computation across all positions and maintaining compatibility with batch decoding.
>
> System-1.5 adopts the same solution. This ensures that early-exited tokens do not disrupt the standard key/value caching mechanism during decoding. Furthermore, System-1.5 retains the same architectural interface as a vanilla Transformer, meaning that other acceleration strategies, such as optimized attention mechanisms, parallel prefill and decode, and hardware-aware scheduling, remain intrinsically compatible.
>
> In summary, System-1.5 is fully compatible with high-performance inference engines and can benefit from their existing optimizations while providing additional speedups through adaptive early exit and step skipping.

---

> > ### Comment · Reviewer_YTmj · 2025-08-05
> >
> > Thank you for your reply. I raised my score to 4.

---

### Note · Authors · 2025-08-13

Dear AC and Reviewers,

We sincerely thank the AC for coordinating the review process and the reviewers for their constructive feedback, which significantly helped us strengthen the technical clarity and rigor of our work.

**Summary of the Original Reviews**

**Major strengths** include the strong motivation and novel dual-shortcut design (YTmj, hpM3, zebB), substantial inference speedups while preserving accuracy (YTmj, zebB, hpM3), clear and modular methodology (zDgS, hpM3), and flexible test-time compute scaling (ws8G).

**Key concerns** include the need for more detailed ablations on shortcut components (YTmj, ws8G), lack of clarity on technical implementation (hpM3), reliance on atom-of-thought–based annotations and their robustness (zDgS, zebB), compatibility with mainstream inference engines and use of smaller-scale models (YTmj), and potential challenges in extending to reinforcement learning settings (zebB).

**Summary of the Rebuttal and Discussion**

We addressed all substantive concerns with targeted clarifications, new experiments, and supporting evidence:

- Clarified the complementary roles of depth and step shortcuts through detailed ablations.

- Scaled experiments to 7B–8B models (LLaMA/Qwen), showing that larger models yield even greater relative speedups.

- Provided detailed explanations of routing behavior, initialization, and training dynamics.

- Demonstrated full compatibility with vLLM and sglang, and showed that training time is comparable to standard CoT fine-tuning.

- Clarified that the ablations noted by Reviewer ws8G were already included in the original submission.

- Added new analyses on training throughput, reasoning difficulty, and pass@k trends under test-time scaling.

- Clarified the dataset construction process and justified the step-level criticality annotation with empirical comparison to token-level supervision.

**Final Resolution**

**Reviewers YTmj and hpM3** raised their scores from 3 to 4.

**Reviewers zDgS and zebB** maintained their positive ratings (4).

**Reviewer ws8G** did not reply during the discussion phase, but we addressed all their points in detail, especially the misinterpretation regarding missing ablations.


We thank the AC and reviewers again for their thoughtful engagement and hope our responses are helpful in the final decision process.

Sincerely,

The Authors of Paper 8563

---

### Decision · Program_Chairs · 2025-09-17

**Decision:**

Accept (poster)

**Comment:**

**(a) Summary**:
This paper proposes *System-1.5 Reasoning*, an adaptive reasoning framework for large language models (LLMs) that reduces inference cost while maintaining reasoning performance. The core idea is to introduce **two dynamic shortcuts** in the latent space:
- **Depth Shortcut (DS):** early exit at intermediate transformer layers to avoid unnecessary computation.
- **Step Shortcut (SS):** skipping reasoning steps by reusing intermediate hidden states.

The framework combines these shortcuts via a routing module that adaptively allocates computation depending on reasoning difficulty. The method is trained in two stages: (1) distillation from a CoT-finetuned teacher into latent reasoning, and (2) shortcut learning guided by reasoning criticality labels derived from atom-of-thought decomposition.

Experiments on reasoning benchmarks (GSM8K, GSM-HARD, StrategyQA, MATH) show that System-1.5 achieves **accuracy comparable to chain-of-thought (CoT)** while providing **substantial speedups (20–30×)**. The approach is also shown to scale effectively to 7B–8B models, outperforming prior latent-space reasoning baselines such as Coconut, iCoT, and CODI.

---

**(b) Strengths**:
1. **Strong motivation and novelty**: Introduces a dual-shortcut mechanism (depth + step) that adaptively balances computation vs. reasoning complexity.
2. **Impressive efficiency gains**: Achieves up to 20–30× inference speedups while retaining accuracy, with benefits growing at larger model scales.
3. **Thorough ablations and clarifications in rebuttal**: Authors provide DS-only vs. SS-only analyses, showing complementary contributions, and analyze training throughput, shortcut routing, and router warm-up behavior.
4. **Compatibility with inference engines**: Demonstrates that the method is compatible with vLLM/sglang and standard KV caching optimizations, ensuring practical relevance.
5. **Robust supervision strategy**: The step-level criticality supervision is shown to outperform noisy token-level heuristics, validating the atom-of-thought decomposition design.
6. **Generalization evidence**: Additional results on OOD datasets (GSM-HARD, MATH) confirm robustness, with consistent efficiency gains even under more challenging settings.

---

**(c) Weaknesses**:
1. **Initial clarity issues**: Reviewers consistently noted that the paper was difficult to follow, especially regarding the computation process, shortcut routing, and training objectives (though rebuttal clarifications helped).
2. **Dataset construction concerns**: Reliance on atom-of-thought decomposition and criticality annotations raised questions about subjectivity and trustworthiness, though authors defended the design with empirical comparisons.
3. **Limited ablations in the original submission**: Key insights (DS vs. SS contribution, curriculum, optimization strategies) were missing from the initial draft and only added during rebuttal.
4. **Framework complexity**: Requires multiple components (teacher, student, criticality pipeline), which may hinder ease of adoption.
5. **Limited experimental diversity at submission time**: The main results focused on smaller models and benchmarks. Larger-scale validation (7B/8B) and additional datasets (MATH) were only introduced during rebuttal.
6. **Writing quality**: Some reviewers (esp. R5) found explanations of equations and algorithmic choices unclear, which could limit accessibility.

---

**(d) Reasons for Acceptance**:
The paper makes a **meaningful and novel contribution** to efficient reasoning in LLMs. Its combination of depth and step shortcuts introduces a new mechanism for dynamic compute allocation in latent space. Despite initial weaknesses in clarity and evaluation scope, the rebuttal provided convincing evidence that System-1.5:
- Achieves **state-of-the-art efficiency–accuracy trade-offs**,
- Scales well to larger models,
- Remains compatible with modern inference engines, and
- Demonstrates robustness across multiple benchmarks.

The efficiency gains (20–30× speedups) are particularly significant given the cost of reasoning with large models, making this work timely and impactful. While the framework is complex, its modularity and practical compatibility mitigate concerns. Overall, the **strengths outweigh the weaknesses**, justifying acceptance.

---

**(e) Summary of Rebuttal and Reviewer Discussion**:
- **Reviewer YTmj (R1):** Initially concerned about missing ablations (DS vs. SS), unclear routing, small model scale, router warm-up, and inference compatibility.
  - **Authors’ response:** Added DS-only and SS-only ablations, clarified routing with thresholds, scaled to 7B/8B models, showed router warm-up curves, and explained vLLM compatibility.
  - **Outcome:** Reviewer raised score to 4, satisfied with clarifications.

- **Reviewer ws8G (R2):** Wanted ablations on curriculum/optimization, training throughput, difficulty-based analysis, and pass@k results.
  - **Authors’ response:** Highlighted existing ablations, provided training time comparisons (System-1.5 nearly as fast as CoT, faster than Coconut), analyzed GSM-HARD vs. GSM8K, and reported pass@k despite limitations in latent reasoning.
  - **Outcome:** Reviewer acknowledged improvements, maintained score of 4.

- **Reviewer zDgS (R3):** Concerned about dataset construction and trustworthiness of shortcut learning labels.
  - **Authors’ response:** Defended atom-of-thought DAGs as structurally coherent supervision, compared against noisy token-level annotations (showing superiority of step-level).
  - **Outcome:** Reviewer remained positive, kept score of 4.

- **Reviewer zebB (R4):** Concerned about dependency on annotation quality, lack of RL compatibility, and framework complexity.
  - **Authors’ response:** Showed robustness to noisy labels, discussed RL extensions via reward modeling, and highlighted modularity + low parameter overhead.
  - **Outcome:** Reviewer accepted rebuttal, maintained positive score.

- **Reviewer hpM3 (R5):** Raised technical clarity issues on equations, shortcut design, and backbone setup. Also questioned GSM-HARD as OOD evaluation.
  - **Authors’ response:** Provided detailed clarifications on computation flow, training vs. inference behavior, DS–SS interaction, consistency loss design, and added MATH experiments.
  - **Outcome:** Reviewer acknowledged better understanding, raised score to 4, but still noted clarity issues and limited scale at submission time.

**Final Decision Considerations**:
All five reviewers converged on **borderline accept (4)** after rebuttal, with improved confidence in the work’s novelty and empirical strength. While weaknesses in clarity, dataset construction, and initial evaluation scope remain, the rebuttal addressed these convincingly. The demonstrated efficiency gains and scalability make this a **worthy contribution for acceptance**.